# GLOBEM Dataset: Multi-Year Datasets for Longitudinal Human Behavior Modeling Generalization

**Xuhai Xu, Han Zhang, Yasaman Sefidgar, Yiyi Ren, Xin Liu, Woosuk Seo, Jennifer Brown**
**Kevin Kuehn, Mike Merrill, Paula Nurius, Shwetak Patel, Tim Althoff**
**Margaret E. Morris, Eve Riskin, Jennifer Mankoff, Anind K. Dey**
University of Washington, Seattle, USA | {xuhaixu,anind}@uw.edu

## Abstract

Recent research has demonstrated the capability of behavior signals captured by smartphones and wearables for longitudinal behavior modeling. However, there is a lack of a comprehensive public dataset that serves as an open testbed for fair comparison among algorithms. Moreover, prior studies mainly evaluate algorithms using data from a single population within a short period, without measuring the cross-dataset generalizability of these algorithms. We present the first multi-year passive sensing datasets, containing over 700 user-years and 497 unique users' data collected from mobile and wearable sensors, together with a wide range of well-being metrics. Our datasets can support multiple cross-dataset evaluations of behavior modeling algorithms' generalizability across different users and years. As a starting point, we provide the benchmark results of 18 algorithms on the task of depression detection. Our results indicate that both prior depression detection algorithms and domain generalization techniques show potential but need further research to achieve adequate cross-dataset generalizability. We envision our multi-year datasets can support the ML community in developing generalizable longitudinal behavior modeling algorithms.

> The GLOBEM website can be found at the-globem.github.io
> Our datasets are available at physionet.org/content/globem
> Our codebase is open-sourced at github.com/UW-EXP/GLOBEM

## 1 Introduction

As machine learning (ML) achieves remarkable success in a wide range of areas, there is a growing need to show real life robustness of ML models through cross-dataset generalizability. Various domain generalization techniques have been proposed to improve model performance when the probability distributions of training data and testing data are different [94, 115]. The majority of existing domain generalization algorithms focus on the tasks of computer vision (CV) [54, 55, 58, 110] and natural language processing (NLP) [10, 27, 42, 93]. Only a few studies have examined domain generalization on time-series data [31, 37, 43], other than short-term human action recognition [57, 114]. However, even this prior research has only investigated time-series data in controlled settings [73] and did not explore domain generalization in longitudinal time-series sensor data in the wild. To build deployable longitudinal time-series systems, it is important to evaluate the model across datasets with different contexts to ensure its generalizability for real-world applications, such as health monitoring [65], medical analysis [62], personalized recommendation [103], and weather prediction [53].

Among various longitudinal sensor streams, smartphones and wearables are arguably one of the most widely available data sources [52]. The advances in mobile technology provide an unprecedented opportunity to capture multiple aspects of daily human behaviors, by collecting continuous sensor streams from these devices [69, 95], together with metrics about health and well-being through self-report or clinical diagnosis as modeling targets. It poses unique challenges compared to traditional

36th Conference on Neural Information Processing Systems (NeurIPS 2022) Track on Datasets and Benchmarks.

time-series classification tasks [43]. First, the data covers a much longer time period, usually across multiple months or years. Second, the nature of longitudinal collection often results in a high data missing rate. Third, the prediction target label is sparse, especially for mental well-being metrics.

In this paper, we focus on longitudinal human behavior modeling, an important multidisciplinary area spanning machine learning, psychology, human-computer interaction, and ubiquitous computing. Researchers have demonstrated the potential of using longitudinal mobile sensing data for behavior modeling in many applications, *e.g.*, detecting physical health issues [65], monitoring mental health status [29], measuring job performance [63], and tracing education outcomes [96]. Most existing research employed off-the-shelf ML algorithms and evaluated them on their private datasets. However, testing a model with new contexts and users is imperative to ensure its practical deployability. To the best of our knowledge, there has been no investigation of the cross-dataset generalizability of longitudinal behavior models, nor an open testbed to evaluate and compare various modeling algorithms. To address this gap, in this paper, we present the first multi-year mobile and wearable sensing datasets to help the ML community explore generalizable longitudinal behavior models.

Our multi-year data collection studies span four years (10 weeks each year, from 2018 to 2021). Each year's dataset includes new and continuing participants. Our datasets contain data collected from 705 person-years (497 unique participants) with diverse racial, ability, and immigrant backgrounds. Each year, they would install a mobile app on their phones and wear a fitness tracker. The app and wearable device passively track multiple sensor streams in the background $24\times7$, including location, phone usage, calls, Bluetooth, physical activity, and sleep behavior. In addition, participants completed weekly short surveys and two comprehensive surveys on health behaviors and symptoms, social well-being, emotional states, mental health, and other metrics. We use the survey data as ground truth for various behavior modeling targets. Our dataset analysis indicates that our datasets capture a wide range of daily human routines, and reveal insights between daily behaviors and important well-being metrics (*e.g.*, depression status). Our datasets can serve as an open testbed for multiple cross-dataset generalization tasks (*e.g.*, same users-different years, different users-different years) to evaluate a behavior modeling algorithm's generalizability and robustness.

As a starting point, we report benchmark results of a behavior modeling task with depression detection as the target, a binary classification task to distinguish whether participants had reported at least mild depressive symptoms using historical mobile and wearable sensing data. We pick depression as a starting point since it is a common and important mental health problem worldwide [90], while we envision our datasets can support other modeling tasks using different labels. We closely re-implement 9 prior depression detection algorithms, 8 recent deep-learning-based domain generalization algorithms, and our recently proposed algorithm, *Reorder* [104]. These 18 algorithms are consolidated on a platform **GLOBEM** (short for **G**eneralization of **LO**ngitudinal **BE**havior **M**odeling) [104]. It has been applied to a multi-institution dataset in [104]. However, this data is not public and does not include pre/post COVID behavioral data. Further, this analysis does not include any benchmarking. We evaluate the generalizability of these algorithms with multiple cross-dataset generalization tasks on the novel four-year datasets, including leave-one-dataset-out, pre/post COVID, and overlapping users across years. Our results indicate that these algorithms can barely generalize across datasets. Although our algorithm *Reorder* has the best overall performance ($\Delta$=15.9% on balanced accuracy over baseline), its advantage is still marginal and far from practical deployability. The community needs more continuing efforts to develop more generalizable behavior modeling algorithms.

**Contributions:** To the best of our knowledge, we present and release the first longitudinal (four-year) mobile and wearable sensing datasets that contain data from over 700 person-years. [1] We report the benchmark results of 18 behavior modeling algorithms for the depression detection task, which indicate the lack of generalizability of all existing algorithms. We envision that our datasets can assist ML researchers' in developing more generalizable longitudinal behavior modeling algorithms and serve as benchmark datasets for longitudinal time-series modeling tasks.

## 2   Background

**Domain Generalization Techniques and Datasets.** A number of domain generalization algorithms have been proposed in the ML community in the past few years. Most of them fall into one of three categories [94]: 1) Data manipulation, which augments or generates data to help the model training

---

[1]Due to the sensitive nature of the dataset, we release our feature-level data with open credentialed access.

Table 1: Comparison of Related Sensor-based Human Behavior Datasets and Research Studies

| | GLOBEM Dataset | StudentLife [4] | CrossCheck [12] | En-Gage [41] | Related Research [20, 97, 101] | Other Human Behavior Datasets WOODS [37] |
|---|---|---|---|---|---|---|
| # of Subjects | 705 (497 unique) | 48 | 34 | 29 | <400 | 9 |
| Time Scale | 3 months×4 years | 10 weeks | 2 years | 4 weeks | Months | Hours×36 devices |
| Open-source | Yes | Yes | Yes | Yes | No | Yes |
| Domain Generalization | Yes | No | No | No | No | Yes |

(*e.g.*, [26, 111]); 2) Representation learning, which focuses on learning generalized representations across domains (*e.g.*, [7, 34, 38]); 3) Learning strategy, which aims to utilize the training procedure to enhance model generalizability (*e.g.*, [30, 108, 86]). Researchers have released multiple datasets such as PACS [56], VLCS [32] and Office-Home [89], and developed cross-dataset benchmark platforms such as DomainBed [46], DeepDG [94], and WILDS [50] to facilitate related studies. However, most existing domain generalization research focuses on the tasks of CV and NLP.

**Generalizable Time-Series Models.** There are fewer studies about model robustness to distribution shift on time-series data [37]. AdaRNN proposes to characterize the temporal distribution shift of signals and reduce the mismatch with an RNN [31]. Godahewa *et al.* provided a dataset archive for general time-series forecasting algorithms evaluation [43]. As for generalizable sensor-based human behavior modeling, some researchers have explored short-term human action recognition [44, 57, 114]. However, these studies primarily rely on data collected in a controlled setting for a short period (minutes to hours) [73, 109]. There is little research focusing on in-the-wild longitudinal human behavior sensor data (months to years) that contains diverse and variable contexts of daily livings.

**Mobile Sensing and Behavior Modeling.** Mobile sensing is one of the most widely available data sources for longitudinal human behavior modeling [21, 40, 52, 67, 68, 79]. Compared to traditional time-series data, mobile sensing data are much longer and uncontrolled (and thus have a high data missing rate [95]). Moreover, the ground truth is usually much more sparse (*e.g.*, self-report mental health measures administered weekly or less frequently [18, 101]). Most existing human behavior modeling algorithms using mobile sensing data are not open-sourced and do not investigate cross-dataset generalization [33, 59, 78, 97, 102, 106]. To date, there are only a few public longitudinal human behavior sensing datasets [4, 12, 41]. Table 1 summarizes and compares them against our multi-year datasets. Existing passive mobile sensing datasets contain fewer than 50 participants and cannot support cross-dataset analysis. They cannot serve as a golden benchmark for future proposed algorithms. We are the first to release multi-year mobile sensing datasets to support the ML community in investigating cross-dataset generalizable behavior modeling algorithms.

## 3 Multi-Year Datasets

We introduce the data collection procedure of our multi-year datasets (Sec. 3.1), together with the details of the survey data (Sec. 3.2) and passive mobile sensing data (Sec. 3.3).

### 3.1 Study Procedure

Our data collection studies were conducted at a Carnegie-classified R-1 university in the United States, inspired by the data collection model proposed in [95]. The study went through an IRB review and approval. Fig. 1 presents the overview of the data collection process.

We recruited undergraduates via emails, flyers, and social posts from 2018 to 2021 [79]. After the first year, previous-year students were invited to join again. The study was conducted during Spring quarter (10 weeks) each year, so the impact of seasonal effects was controlled. Participants received up to $245 in compensation based on their compliance each year. S.A.1 provides more study details.

The four datasets (DS1 to DS4) have 155, 218, 137, and 195 participants (705 person-years overall, and 497 unique people). We intentionally oversampled minoritized groups to make our datasets more representative. Our datasets have a high representation of females (58.9%), immigrants (24.2%), first-generations (38.2%), and people with disability (9.1%), and have a wide coverage of races, with Asian (53.9%) and White (31.9%) being dominant (Hispanic/Latino 7.4%, Black/African American 3.3%). S.A.2 summarizes the demographics and S.A.4 discusses the intrinsic bias.

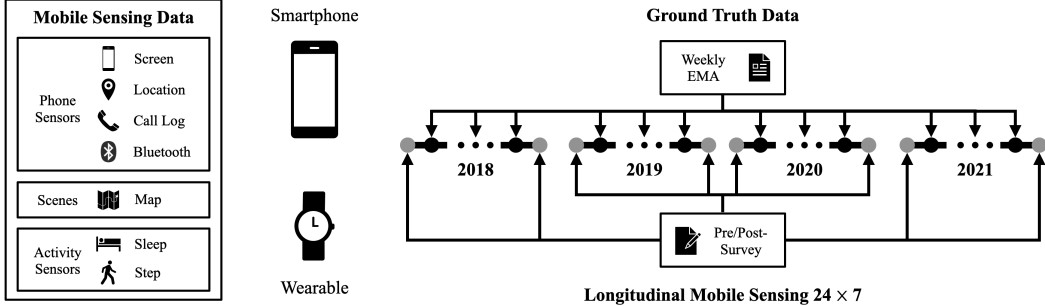

Figure 1: Overview of Longitudinal Passive Sensing Data Collection Studies. Each year's study lasted a 10-week academic quarter.

## 3.2 Survey Data

We collected survey data at multiple stages of the study. We delivered extensive surveys before the start and at the end of the study (pre/post surveys) and weekly Ecological Momentary Assessment (EMA) surveys during the study to collect in-the-moment self-report data. All surveys consist of well-established and validated questionnaires to ensure data quality.

Our pre/post surveys include a number of questionnaires to cover various aspects of life, including 1) personality (BFI-10, The Big-Five Inventory-10 [75]), 2) physical health (CHIPS, Cohen-Hoberman Inventory of Physical Symptoms [23]), 3) mental well-being (*e.g.*, BDI-II, Beck Depression Inventory-II [11]; ERQ, Emotion Regulation Questionnaire [45] ), and 4) social well-being (*e.g.*, Sense of Social and Academic Fit Scale [92]; EDS, Everyday Discrimination Scale [5, 100]).

Our EMA surveys focus on capturing participants' recent sense of their mental health, including PHQ-4, Patient Health Questionnaire 4 [6, 51]; PSS-4, Perceived Stress Scale 4 [1, 24]; and PANAS, Positive and Negative Affect Schedule [2, 99]. S.A.6 lists details of each questionnaire.

As an initial step of model generalizability evaluation, we focus on detecting mental health concerns. We employ BDI-II (post) and PHQ-4 (EMA) as the ground truth. Both are screening tools for further inquiry of clinical depression or anxiety diagnosis. We focus on a binary classification problem to distinguish whether participants' scores indicate at least mild mental health concerns (*i.e.*, PHQ-4 > 2, BDI-II > 13)[2]. We use "depression detection" as shorthand for detecting this group of mental health concerns in the paper. The average number of depression labels is 11.6±2.6 per person. Fig. 2 sum-

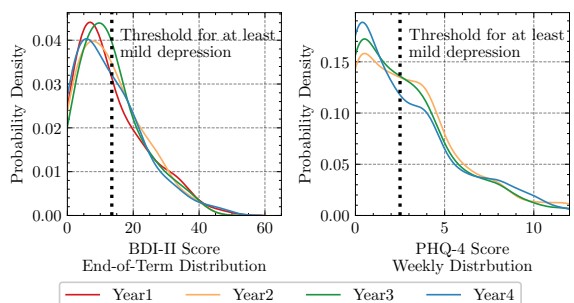

Figure 2: The Distribution of Label Scores for End-of-Term (BDI-II) and Weekly Depression Scales (PHQ-4).

marizes the distribution of survey scores across four datasets. The percentage of reports with at least mild depression is 39.8±2.7% for BDI-II and 47.4±2.8% for PHQ-4.

## 3.3 Sensor Data

We developed a mobile app using the AWARE Framework [35] that continuously collects location, phone usage (screen status), Bluetooth scans, and call logs. The app is compatible with both the iOS and Android platforms. Participants installed the app on smartphones and left it running in the background. In addition, we provided Fitbits to collect their physical activities and sleep behaviors. The mobile app and wearable passively collected sensor data 24×7 during the study. The average number of days per person per year is 77.5±8.9 among the four datasets.

---

[2]PHQ-4 contains two sub-scales for depression and anxiety. We use the overall PHQ-4 score, allowing us to combine PHQ-4 and BDI-II as both use a 4-level health concern categorization (normal, mild, moderate, and severe). For a stricter focus on depression, we recommend using the depression sub-scale of PHQ-4. Moreover, since DS1 did not have PHQ-4, we used another questionnaire as a substitute. Please refer to S.B.1 for details.

We utilize RAPIDS [3, 88], an open-source platform that provides a Reproducible Analysis Pipeline for Data Streams. It supports feature extraction from data collected via multiple mobile and wearable devices with various time windows. S.A.7 lists feature details and potential limitations.

**Data Type: Location.** We incorporate all features in RAPIDS-Location, which includes location variance, location entropy, travel distance, *etc.* In addition, we also added more features (duration of staying) for specific points of interest, including places for living, study, exercise, and relaxation.

**Data Type: Phone Usage.** We include all features in RAPIDS-Screen that cover the statistics of unlocking episodes (count, sum, mean, std, max, min). We further contextualize these features at different locations (home and study places) to capture fine-grained phone usage behaviors.

**Data Type: Bluetooth.** We use all features from RAPIDS-Bluetooth, including the number of scans of participants' own devices and others' devices, as well as the unique count of these devices.

**Data Type: Call.** We employ features from RAPIDS-Call that cover the statistics of incoming/outgoing calls' duration (count, sum, mean, std, max, min, entropy), and the count of missed calls.

**Data Type: Physical Activity.** We utilize physical activity features from RAPIDS-Fitbit-Steps. They include both high-level features (number of steps, duration of being active), and low-level features about the statistics of active or sedentary episodes (mean, std, max, min).

**Data Type: Sleep.** We leverage sleep-related features from RAPIDS-Fitbit-Sleep, including high-level summary features (total duration of being asleep or in bed), and low-level features about the statistics (count, mean, max, min) of episodes of being asleep, restless, and awake during the sleep.

**Feature Time Range.** Research has found that people tend to have distinctive behavior patterns during different times of the day [22], or accumulate their behavior routines through a period of days [18]. Thus we incorporate different time ranges during feature extraction, including four epochs of a day (split at 6 am, 12 pm, 6 pm, and 12 am), the whole day, the past one/two weeks. It is worth noting that all features are calculated every day for each user, forming a long daily feature vector.

**Post-processing.** After feature extraction, we further conducted a few post-processing steps to provide a comprehensive feature set: 1) Feature normalization: We add all features' normalized version based on each individual's distribution: subtracting the median and scaling with the 5-95 quantile range on each individual; 2) Feature discretization: A few modeling algorithms may benefit from using categorical levels instead of raw feature values (*e.g.*, [101]), thus we also add all features' 3-level discretized version (split by the one/two/three third percentile within each individual's data).

Missing data is inevitable due to various reasons, such as low battery, data transfer loss, and sensor permission withdrawal. For example, the average missing rate for location features is 14.5±4.0%. Please find more details about the missing rates of different features in S.A.7. We omit missing values during analysis and use a median-based imputation when necessary.

## 4 Dataset Analysis

Our multi-year datasets capture various aspects of participants' daily routines (Sec. 4.1), and reveal important insights into the relationship between daily behaviors and mental health metrics (Sec. 4.2). Meanwhile, the datasets also demonstrate potential domain generalization challenges (Sec. 4.3).

### 4.1 Data Distribution

Each year's dataset covers a period of 10 weeks. Fig. 3 visualizes the daily value of three representative features across all years. Since the period of DS3 collection began right after the national lockdown (Mar to Jun, 2020), the impact of COVID is clearly reflected in the differences between DS1&2 *vs.* DS3&4 on the mobility-related features [70, 112, 113]. For example, the daily step count of DS3&4 drops by nearly half. Meanwhile, we can observe a recovery trend when comparing DS3 and DS4, as indicated by the increased travel distance and step counts. Interestingly, the travel distance in DS4 is close to DS1&2, while the step count is still much lower. This may suggest that participants used commuting methods other than walking even after cities were re-opened. Moreover, the weekly routine cycle is salient in all years. The daily travel distance significantly increases on weekends (mostly on Saturdays), while the walking step counts drop. Further, participants tended to leverage weekends to catch up on sleep, as shown by the peak in-bed duration around weekends.

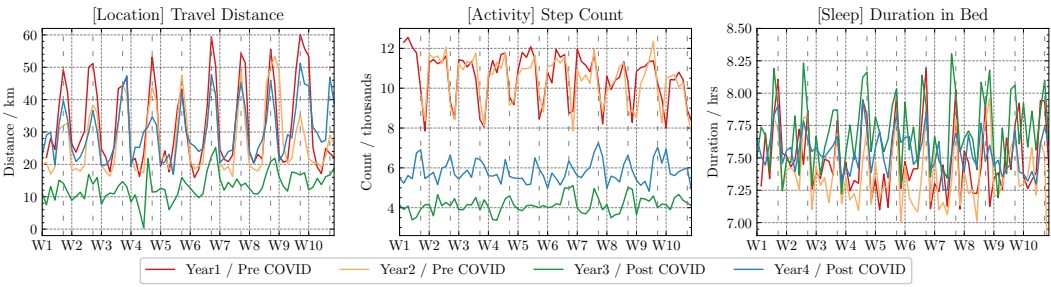

Figure 3: Time-series of Example Features. Grids split weeks. Dashed lines split weekdays/weekends.

We further compare the probability density function (PDF) shapes of features across years in Fig. 4, using an example from each sensor type. We observe the distinction between DS1&2 *vs.* DS3&4. This again reveals the impact of the pandemic. Other than the similar observations from Fig. 3, we further find that participants visited fewer places, spent more time on smartphones, had longer phone call durations, and joined fewer social activities (as indicated by Bluetooth as proxy). These observations indicate that our datasets capture different aspects of daily routines and routine changes.

In addition, despite the similarity in some features' PDFs, each DS has its own unique feature distribution. For example, DS2 has a bimodal distribution on the number of frequent locations, while others' are unimodal. DS3's sleep duration has a slight distribution shift towards the right (*i.e.*, participants tended to sleep more right after the lockdown). These distribution shifts suggest challenges for cross-dataset generalization of longitudinal modeling (see Sec. 4.3 for more details).

## 4.2 Correlation Analysis

Our datasets not only reflect participants' daily routines, but also capture the relationship between daily behaviors and well-being metrics. We use depression as an example for correlation analysis.

We compute Spearman correlation coefficients $\rho$ between every feature and the depression label in each dataset. Figure 5 shows top features from each type with significant $\rho$s ($p < 0.05$) and the same directions in all datasets. There are some interesting findings. For example, the past two weeks' sleep duration and count of screen unlock episodes at night have the strongest correlation ($|\rho| > 0.1$). Shorter sleep duration and more screen usage are associated with higher depression scores. These are supported by the psychology and psychiatry literature, which suggests disturbed sleep patterns and lack of focus are common depressive symptoms [8, 83, 85]. Moreover, other features indicate that participants with higher depression scores tended to have less physical activity and lower mobility,

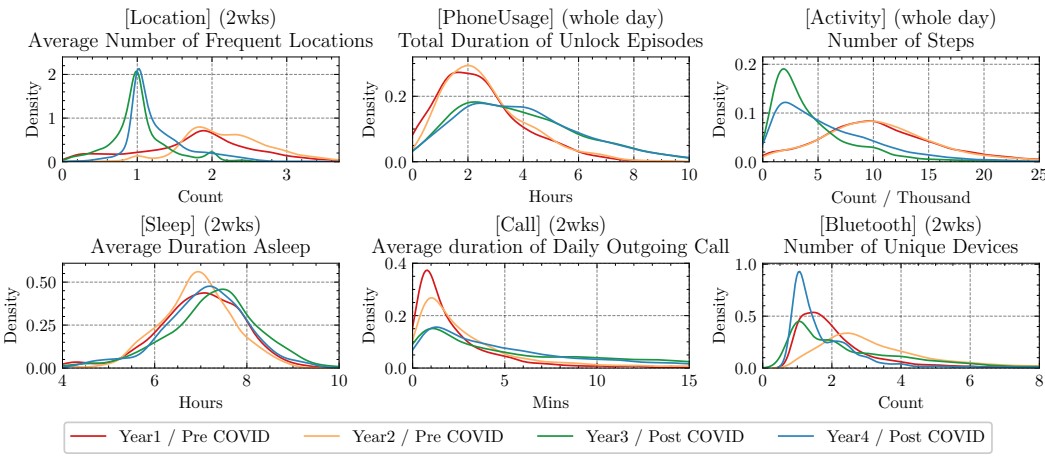

Figure 4: Distribution of Example Features from Each Sensor Types.

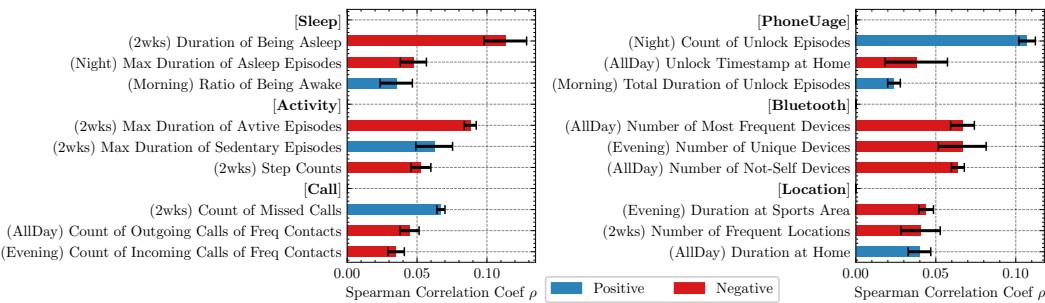

Figure 5: Correlation Analysis of Representative Feature Value and Depression Labels

spend more time at home, and engage in less social communication. These observations reflect a sign of diminished interest in other activities, another common symptom of depression [17, 76].

### 4.3 Domain Classification

To quantify the differences among datasets, we first conduct a "Name-The-Dataset" task on the four datasets [84], treating each dataset as a domain. We split the users 80%/20% into training/testing set, and use daily features as the input. We use a portion of users in the training data to train a small Random Forest (RF, n=10, max depth=3) to classify which dataset a data belongs to (*i.e.*, four-class classification). The left side of Fig. 6a shows the results. With 1/10/100 users (0.2%/2%/20% of the training set), the model can achieve an accuracy of 62.3%/84.2%/91.1%, which indicates that behavior features from different DS have distinguishing distributions. We also repeated the training with normalized features, as shown in the right side of Fig. 6a. The normalization can reduce the distribution shift, especially for DS1 and DS4, but the distinction between datasets still persists.

We further conduct a "Distinguish-The-Person" task, with each person-year as a domain. This time the 80%/20% split is performed on each person's data. We train another RF (n=10, max leaf num=2K) to classify which person a data belongs to (*i.e.*, 705-class classification). This is a more challenging task, but the model still achieves an accuracy of 7.7%/26.2%/46.3% when using 1/10/50 days of data from each participant (1.3%/13%65% of the training set). Meanwhile, the normalization does not significantly diminish the effect of distribution shift in this task, as shown in Fig. 6b. These results indicate that there exist significant distribution shifts among datasets and individuals. Our benchmark results in Sec. 5 demonstrate the challenges for domain generalization on behavior modeling tasks.

## 5 Benchmark

There is a growing body of research showing that passive sensing data from everyday devices can capture daily behavior signals related to depressive symptoms [9, 12, 18, 52], which has attracted increasing attention from various communities. Therefore, we use depression detection as the main task to benchmark our multi-year datasets. We envision the platform can be extended to other behavior modeling tasks using different ground truth labels.

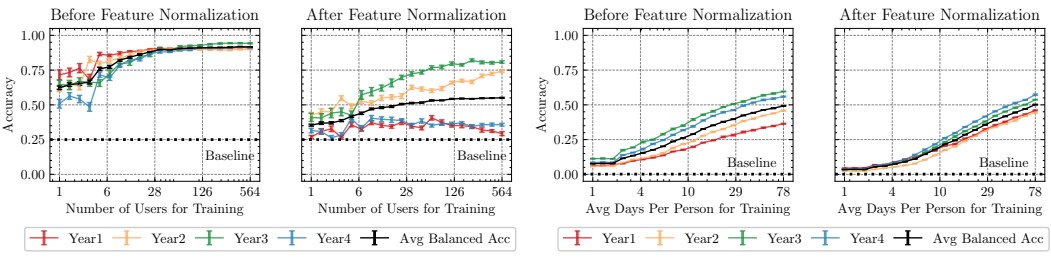

(a) Name-The-Dataset (n=10, max depth=3)  (b) Distinguish-The-Person (n=10, max leaf num = 2K))

Figure 6: Performance of Domain Classification with Simple Random Forest Models.

## 5.1 Data Preparation

The raw data format is a time-series feature-vector for each participant, with a short list of labels on certain dates. Since data length varies across participants, we slice the feature sequence based on labels to construct consistent inputs. Given a label collected on a date, we collect a feature matrix of the past four weeks to cover behavior trajectory history [18, 60]. After the slicing, every data point corresponds to one label and an input feature matrix with the same shape (28 days $\times$ feature number).

## 5.2 Behavior Modeling Algorithms

GLOBEM [104] closely re-implements 9 prior depression detection algorithms and 9 deep-learning domain generalization algorithms for consistent evaluation. The details of the algorithms, hyperparameters, and model training are described in S.B.2 and [104].

**Depression Detection Algorithms.** Researchers in the ubiquitous computing community have proposed a range of algorithms that use passive mobile sensing for depression detection. Due to the limited size of these datasets, these methods mostly aggregate a subset of features within certain time ranges and train off-the-shelf traditional ML models: 1) *Canzian et al.* [18]: uses some location features to train an SVM; 2) *Saeb et al.* [78]: uses a subset of location and screen features to train a logistic regression model; 3) *Farhan et al.* [33]: uses location and physical activity features to train an SVM; 4) *Wahle et al.* [91]: uses features from several sensors to build SVM and Random Forest models; 5) *Lu et al.* [60]: uses multiple sensor features to build multi-task learning models combining linear and logistic regression; 6) *Wang et al.* [97]: calculates the average and slope of the past two weeks of behavior features, and builds a lasso-regularized logistic regression model; 7) *Xu et al.-I* [101]: applies association rule mining on behavior features to extract contextually filtered features to build an Adaboost model; 8) *Xu et al.-P* [102]: uses a collaborative-filtering-based model with the square of Pearson correlation coefficient as the weights; 9) *Chikersal et al.* [20]: calculates breakpoint and slope of multiple features, trains a gradient boosting model for each sensor, and combines them with an Adaboost model.

**Domain Generalization Algorithms.** These techniques use the same set of features (*i.e.*, the same feature matrix) as the input. We pick representative ones to cover major directions of domain generalization [94]: 1) *ERM* (Empirical Risk Minimization) [87]. We implement multiple architectures with ERM: *ERM-1D-CNN*, *ERM-2D-CNN*, *ERM-LSTM*, *ERM-Transformer*; 2) *Mixup* [111]; 3) *IRM* (Invariant Risk Minimization) [7]; 4) *DANN* (Domain-Adversarial Neural Network) [38]. We test both using dataset as a domain (*DANN-Dataset as Domain*), and person as a domain (*DANN-P*); 5) *CSD* (Common Specific Decomposition) [72]. Similarly, we also test *CSD-D* and *CSD-P*; 6) *MLDG* (Meta-Learning for Domain Generalization) [56], with *MLDG-D*, and *MLDG-P*; 7) *MASF* (Model-Agnostic Learning of Semantic Features) [30], with *MASF-D*, and *MASF-P*; 8) *Siamese Network* [49]; 9) *Reorder* [104], a self-supervised learning-based algorithm that leverages order reconstruction of a shuffled sequence as the pre-text task. Algorithms from 2-9 use the same 1D-CNN as the backbone.

## 5.3 Experiment Setup

We experiment with multiple setups to evaluate algorithm performance: 1) Users Past/Future within One Dataset, a simple setup that uses the first 80% of every user's data as the training set, and the remaining 20% as the testing set in each DS. 2) Leave-One-Dataset-Out, a cross-dataset setup that uses three DS as the training set, and the other as the testing set. 3) Pre/Post-COVID, another cross-dataset setup to measure the effect of the pandemic, using DS1&2 (before COVID) as the training set and DS3&4 (after COVID) as the testing set, and then swapping the two sides. 4) Overlapping Users across Datasets, a cross-dataset setup that only focuses on overlapping users in multiple datasets to measure the time effect, which trains a model with overlapping users from one dataset, and tests it on overlapping users from other datasets. We employ balanced accuracy (the average of sensitivity and specificity) as the metric, as it has been shown to be more robust to class-imbalance [15].

## 5.4 Model Performance

Tab. 2 summarizes the results of all algorithms in different setups. We highlight the important observations. No single depression detection algorithm stands out over most tasks. *Xu et al.-I* is the best for

Table 2: Model Balanced Accuracy of Depression Detection under Different Setups.

| Category | Model | Single Dataset | Cross Dataset | | |
|---|---|---|---|---|---|
| | | Past/Future | Leave-One-DS-Out | Pre/Post-COVID | Overlapping Users |
| Baseline | Majority | 0.500±0.000 | 0.500±0.000 | 0.500±0.000 | 0.500±0.000 |
| Prior Depression Detection Model | Canzian *et al.* [18] | 0.536±0.026 | 0.498±0.006 | 0.497±0.003 | 0.496±0.031 |
| | Saeb *et al.* [78] | 0.557±0.020 | 0.536±0.008 | 0.519±0.004 | **0.565±0.039** |
| | Farhan *et al.* [33] | 0.562±0.021 | 0.506±0.007 | 0.500±0.019 | 0.480±0.013 |
| | Wahle *et al.* [91] | 0.598±0.020 | 0.524±0.011 | 0.526±0.003 | 0.512±0.013 |
| | Lu *et al.* [60] | 0.550±0.024 | 0.531±0.011 | 0.505±0.007 | 0.508±0.022 |
| | Wang *et al.* [97] | 0.530±0.020 | 0.521±0.007 | 0.524±0.010 | 0.532±0.028 |
| | Xu *et al.*-I [101] | **0.691±0.018** | 0.502±0.012 | 0.519±0.019 | 0.494±0.013 |
| | Xu *et al.*-P [102] | 0.600±0.007 | 0.502±0.006 | 0.508±0.003 | 0.544±0.009 |
| | Chikersal *et al.* [20] | 0.649±0.016 | **0.536±0.002** | **0.528±0.024** | 0.545±0.032 |
| Recent Domain Generalization Model | ERM-1dCNN [87] | 0.568±0.006 | 0.510±0.008 | 0.514±0.006 | 0.534±0.007 |
| | ERM-2dCNN [87] | 0.533±0.013 | 0.510±0.006 | 0.504±0.006 | 0.520±0.011 |
| | ERM-LSTM [87] | 0.565±0.019 | 0.512±0.006 | 0.512±0.003 | 0.525±0.020 |
| | ERM-Transformer [87] | 0.584±0.013 | 0.509±0.008 | 0.512±0.016 | 0.506±0.005 |
| | ERM-Mixup [111] | 0.568±0.006 | 0.501±0.008 | 0.507±0.004 | 0.534±0.007 |
| | IRM [7] | 0.573±0.016 | 0.506±0.006 | 0.499±0.000 | 0.508±0.015 |
| | DANN-D [39] | 0.526±0.016 | 0.514±0.004 | 0.514±0.000 | 0.482±0.013 |
| | DANN-P [39] | 0.502±0.002 | 0.500±0.000 | 0.500±0.000 | 0.486±0.017 |
| | CSD-D [72] | 0.562±0.022 | 0.521±0.002 | 0.512±0.006 | 0.517±0.025 |
| | CSD-P [72] | 0.542±0.010 | 0.511±0.006 | 0.516±0.000 | 0.515±0.028 |
| | MLDG-D [56] | 0.522±0.013 | 0.511±0.006 | 0.495±0.004 | 0.519±0.014 |
| | MLDG-P [56] | 0.508±0.011 | 0.510±0.003 | 0.500±0.003 | 0.511±0.016 |
| | MASF-D [30] | 0.505±0.006 | 0.505±0.001 | 0.504±0.007 | 0.532±0.015 |
| | MASF-P [30] | 0.495±0.007 | 0.505±0.004 | 0.509±0.011 | 0.530±0.011 |
| | Siamese Network [49] | 0.545±0.025 | 0.509±0.010 | 0.515±0.002 | 0.527±0.031 |
| | Reorder [104] | **0.626±0.009** | **0.547±0.008** | **0.525±0.003** | **0.573±0.030** |

the single-dataset setup ($\Delta$=38.2% over the naive majority baseline), and *Chikersal et al.* has the overall best performance on cross-dataset setups ($\Delta$=7.2%). Among domain generalization algorithms, *Reorder* has the best overall performance ($\Delta$=25.2% for single-dataset, $\Delta$=9.7% for cross-dataset). Comparing each setup's top algorithm between the two categories, the best depression detection algorithms are better at the single-dataset task ($\Delta$=10.4%), while the best domain generalization algorithms are better at cross-dataset tasks ($\Delta$=2.3%), which shows better generalizability.

More importantly, we observe a significant performance drop from the single dataset task to the three cross-dataset tasks ($\Delta$=7.6±6.7%), especially for algorithms that have good single-dataset performance (*e.g.*, *Xu et al.-I* $\Delta$=26.9%, *Reorder* $\Delta$=12.4%). Current algorithms' cross-dataset generalizability is still far from satisfactory for real-life deployment.

### 5.5   Ethical Consideration

The purpose of using widely available passive sensing data for human behavior modeling, especially for mental health issue detection (*e.g.*, depression in our task), may be arguable. Current research studies assume a positive goal of applying such modeling techniques to support early diagnosis and future adaptive intervention design [105]. But we may need careful regulations on practitioners and stakeholders to avoid negative uses, such as selling under-verified products/medications, or providing mental health support services that are not well-suited to individuals.

Privacy is another major ethical concern of our data collection studies. We strictly follow our IRB's rules for anonymizing participants' data. Since some sensitive sensor data (*e.g.*, location) can disclose identities, we only release feature-level data under credentialing to protect against privacy leakage. Please refer to S.C for our data sharing and maintenance plan. Further, our datasets have diverse yet unbalanced groups (*e.g.*, racial groups), which could introduce bias in model training against underrepresented minorities. S.A.4 discusses more aspects of potential intrinsic bias.

## 6   Discussion

**Insights from Our Datasets.** Our datasets cover over 700 person-years across four years from diverse user groups. The analysis in Sec. 4.1 indicates that the datasets capture various aspects of life experiences, including general behavior patterns, a weekly routine cycle, the impact of COVID,

and the gradual recovery after COVID. Moreover, Sec. 4.2 uses depression as the target and reveals that some behavior features have a consistent correlation across multiple datasets with the scores of depression scales (*e.g.*, less physical movement, more disturbed sleep patterns, less social activities), which are supported by literature in psychology and psychiatry [8, 76, 83]. Please refer to Sec. A.5 for additional correlation analysis between pre- and post-COVID periods. Compared to most prior studies using a single dataset (*e.g.*, [95, 101]), our findings have stronger validity and credibility.

**Lack of Generalizability of Existing Algorithms.** Despite some similarity across datasets, Sec. 4.3 indicates distribution shifts across datasets and individuals. To some extent, this is expected due to the different societal contexts each year and the uniqueness of each person's behavior patterns [102]. However, our benchmark results in Sec. 5.4 demonstrate that both prior depression detection algorithms and recent domain generalization techniques suffer from overfitting and cannot generalize well across datasets. This may be explained by the fact that most domain generalization algorithms we implemented were proposed for CV/NLP tasks, and were not designed for the longitudinal modeling tasks. Although *Reorder* achieves the best generalization performance, it is still far from practical deployability. These results indicate that further advances in generalizability are much needed in the area of longitudinal behavior modeling.

**Prospective Directions to Improve Model Generalizability.** There are two major challenges of generalizability, which illuminates two potential directions to improve model performance: behavior change of an individual across time, and behavior differences between individuals. Compared to other cross-dataset setups, the setup of overlapping users has a relative performance advantage (see Table 2). This indicates that addressing temporal shifts along a single individual's longitudinal behavior could be a relatively easier task. Some recent algorithms such as AdaRNN [31] are designed to address this challenge and are worth testing. As for the individual difference, *Reorder* indicates that leveraging a pre-text shuffling and reordering task may push the model to learn more generalizable representations. This suggests that designing more pre-text tasks that can capture the nature of human behavior could be another future direction, *e.g.*, a task to predict the immediate next behavior feature value (analogous to the pre-text task of BERT [27]).

**Other Potential Behavior Modeling Tasks.** Our experiments and benchmark results focus on the depression detection task. Our datasets contain rich ground truth labels that can support a wide range of behavior modeling tasks. For frequent weekly prediction tasks, our datasets also have labels of participants' stress level (PSS-4) and emotions (PANAS). These labels can enable longitudinal stress detection or emotion monitoring tasks, which can be complementary to existing research using short-term physiological sensing data such as PPG and GSR signals (*e.g.*, [61, 66]). Moreover, our datasets can be used for other behavior modeling tasks with less frequent labels, such as personality prediction [98] (BFI10), social loneliness evaluation [29] (UCLA, Social Fit), discrimination event detection [79] (EDS), *etc.* Please refer to S.A.6 for a comprehensive list of survey data we collected, which provide the community with the potential to explore diverse modeling tasks.

**Limitations & Future Work.** There are some limitations that can be addressed in future work, such as more diverse populations beyond young adults, more sensor signals such as HRV and SpO2 measures from wearables, and better missing data processing methods. It is worth noting that the validity of using self-report for depression measures and other mental health classifications is still debated [36], creating inherent challenges for model development. However, more valid ground truth such as clinical diagnosis are harder to obtain and less frequent. In addition, sensor error across phone and wearable models may introduce additional noise to the datasets [82]. Also, more advanced data imputation techniques, recent adaptive time-series algorithms, and other modeling targets besides depression can be evaluated on our datasets. These behavior models may shed light on the future work of developing intelligent, just-in-time adaptive intervention techniques [71, 107].

# 7 Conclusion

We release the first multi-year longitudinal mobile sensing datasets with multiple sensor streams and various well-being metrics. Our analysis indicates that the datasets capture a range of daily routines, revealing insights between daily behaviors and important well-being metrics such as depression status. Our benchmark results reveal the challenge and the opportunity for the ML community to develop generalizable longitudinal behavior modeling algorithms. We also envision our datasets serving as a gold-standard benchmark for future machine learning research in longitudinal time-series data for human behavior modeling.

## Acknowledgments and Disclosure of Funding

Our multi-year data collection study closely followed a sister study at Carnegie Mellon University (CMU). We acknowledge all efforts from CMU Study Team to provide important starting and reference materials. Moreover, our studies were greatly inspired by StudentLife researchers from Dartmouth College.

Our studies were supported by the University of Washington (including the Paul G. Allen School of Computer Science and Engineering; Department of Electrical and Computer Engineering; Population Health; Addictions, Drug and Alcohol Institute; and the Center for Research and Education on Accessible Technology and Experiences); the National Science Foundation (EDA-2009977, CHS-2016365, CHS-1941537, IIS1816687 and IIS7974751), the National Institute on Disability, Independent Living and Rehabilitation Research (90DPGE0003-01), Samsung Research America, and Google.

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

# A    Additional Study Details

## A.1    Study Documents

We provide a few important documents used in our data collection studies. Please find these files in the supplementary folder:

1. University IRB Approval Letter: The letter from University IRB to approve our studies.
2. Consent Form: The form to be signed to participants before joining the study.
3. Compensation Structure: Participants will earn up to $245 based on their participation compliance.
4. Participant instruction (iOS version, and Android version): Slide decks to guide participants through the app installation and Fitbit setup during the on-boarding.

## A.2    Study Demographics

Table 3: Basic Study Information and Participant Demographics of Four Datasets. Participants with less than 2 weekly EMAs or less than a 25% of their sensor data (*i.e.*, missing rate $> 75\%$) were excluded from the dataset. In the depression row, the percent indicates the portion of participants having at least mild depressive symptoms based on the corresponding questionnaires. Gender acronym - F: Female, M: Male, NB: Non-binary. Generation acronym - Im: Immigrant (born in another country), 1stG: First generation (parents immigrated to the US), 2ndG: Second generation (grandparents immigrated to the US), 3rdG: Third generation (great grandparents or further back immigrated to the US), NA: Prefer not to respond. Racial acronym - A: Asian, B: Black or African American, H: Hispanic or Latino, N: American Indian/Alaska Native, PI: Pacific Islander, W: White, NA: Did not report. & is used when participants reported more than one races.

|  | Year1 - DS1 | Year2 - DS2 | Year3 - DS3 | Year4 - DS4 |
|---|---|---|---|---|
| **Participants** | ● Total: 155
● Gender: F 107, M 48
● Generation: Im 34, 1stG 53, 2ndG 11, 3rdG 57
● Disability: 5
● Race: A 82, B 5, H 9, N 4, PI 3, W 50, A&PI 2 | - Total: 218
● Gender: F 111, M 107
● Generation: Im 54, 1stG 75, 2ndG 18, 3rdG 63, NA 8
● Disability: 21
● Race: A 102, B 6, H 10, N 2, PI 1, W 70, A&B 1, A&W 16, H&W 2, B&W 2, A&H&W 1, B&H&W 1, H&N&W 1, NA 3
● Overlap: 23 in Year1 | - Total: 137
● Gender: F 75, M 61, NB 1
● Generation: Im 35, 1stG 52, 2ndG 8, 3rdG 40, NA 2
● Disability: 22
● Race: A 74, B 3, H 8, PI 3, W 40, A&W 6, B&H&W 1, NA 2
● Overlap: 19 in Year1&2, 4/47 in Year1/2 | - Total: 195
● Gender: F 122, M 67, NB 6
● Generation: Im 48, 1stG 89, 2ndG 13, 3rdG 42, NA 3
● Disability: 16
● Race: A 104, B 4, H 18, N 1, PI 2, W 48, A&W 13, H&W 2, NA 3
● Overlap: 19 in Year1&2&3, 4 in Year1&2, 4 in Year1&3, 47 in Year2&3, 2/19/20 in Year1/2/3 |
| **Survey** | ● Pre/post: UCLA, SocialFit, 2-Way SSS, PSS, ERQ, BRS, CHIPS, STAI, CES-D, BDI2, MAAS, BFI10, Brief-COPE, GQ, FSPWB, EDS, CEDH, B-YAACQ
● Weekly EMA: PHQ-4, PSS-4, PANAS | | | |
| **Depression** | ● Weekly: Depression & Affect (45.5%)
● End-term: BDI-II (35.4%) | ● Weekly: PHQ-4 (52.1%)
● End-term: BDI-II (42.9%) | ● Weekly: PHQ-4 (46.9%)
● End-term: BDI-II (40.7%) | ● Weekly: PHQ-4 (45.0%)
● End-term: BDI-II (40.2%) |
| **Sensor** | ● Smartphone: Location, Phone Usage, Call, Bluetooth
● Wearable: Physical Activity, Sleep | | | |

## A.3    Study Hardware and Setup

Our smartphone data collection app is compatible with both iOS and Android platforms. Therefore, we did not have limits on participants' devices. Before each year's study, we tested our app on multiple smartphone brands to ensure its compatibility, robustness, and data collection quality. However, problems such as smartphone battery drain, software crashes, and data uploading error are inevitable during the study. Thus, we developed a study dashboard to monitor the condition of data collection during the study, and our study team would reach out to help participants solve software or hardware when necessary.

Figure 7 presents a screenshot of the app. The interface is consistent on both platforms. Users can click 1) the "Save" button to manually trigger data uploading, 2) the "Open Survey" button to manually enter the survey if that's within designated time windows, (note that participants usually received EMAs through notifications), and 3) the "Refresh Fitbit Token" for Fitbit data access update.

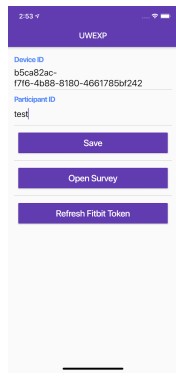

Figure 7: App Screenshot

As for wearables, we used two models of Fitbit (Flex2 for Year 1&2 and Inspire2 for Year 3&4). Both models support reliable physical activity and sleep behavior tracking, but not others (*e.g.*, heart rate tracking). Our internal team also tested and compared the two Fitbit models' tracking accuracy and did not observe significant difference.

## A.4 Study Intrinsic Bias

We discuss some potential intrinsic bias in our datasets. For example:

1. Recruitment Bias: Only a portion of students who received our emails or social media posts would participate in our study, which could only represent a subset of the general population.
2. Gender Group Bias: Our studies intentionally over-sample females, which could involve bias towards the female group.
3. Generation Group Bias: Our studies intentionally over-sample immigrants and first-generation participants, which could involve bias against other generation groups.
4. Racial Group Bias: Asian and White are two dominant racial groups in our studies, while other racial groups are less represented. This could introduce racial bias.
5. Health Group Bias: Some health conditions would impact participants' compliance. For example, participants with severe depressive symptoms may stop responding to surveys or even charging their phones, which would introduce bias into the missing data rate.
6. Device Bias: Although our data collection app is compatible with both iOS and Android platforms, the differences between OS systems and smartphone models may introduce bias into the dataset.

We look forward to future exploration of these different aspects of intrinsic bias.

## A.5 Additional Correlation Analysis

In addition to identifying features that have a consistent correlation with the depression label across all years' datasets (see Figure 5), we are also interested in the features that have opposite correlation directions between pre-COVID and post-COVID periods. We followed a similar procedure as Sec. 4.2 to find features that have a consistent and significant correlation direction within two years (DS1&2, or DS3&4) but an opposite direction between pre- and post- COVID datasets. Figure 8 shows one representative feature from each data type.

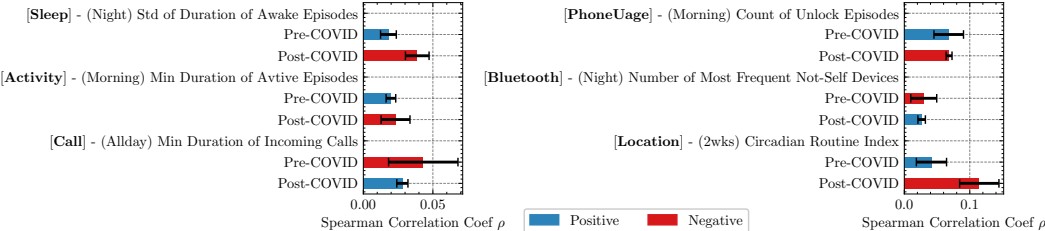

Figure 8: Correlation Analysis of Representative Contrasting Feature Value and Depression Labels

There are some interesting findings, especially when compared against Figure 5. For example, Figure 5 indicates that generally more frequent and longer smartphone usage is positively correlated with depression labels. However, in the morning time, this finding only holds before COVID. After the outbreak of COVID, frequent usage of a smartphone becomes negatively correlated with depression. This may be explained by the fact that the smartphone becomes a necessary tool for all kinds of daily routines when people are locked at home, which could overturn the correlation direction as participants with depression may tend to lose interest in general activities [8]. We look forward to more analysis and insights from future researchers.

## A.6 Survey Details

We list out the survey names and short descriptions used in our study. Please find specific question items in the supplementary folder.

Table 4: Description of Survey Scales

| Scale Name & Abbreviation | Short Description | Scoring Range | Year | Collection Time |
|---|---|---|---|---|
| UCLA [77] Short-form UCLA Loneliness Scale | A 10-item scale measuring one's subjective feelings of loneliness as well as social isolation. Items 2, 6, 10, 11, 13, 14, 16, 18, 19, and 20 of the original scale are included in the short form. Higher values indicate more subjective loneliness. | 10 - 40 | 1,2,3,4 | pre, post |
| Social Fit [92] Sense of Social and Academic Fit Scale | A 17-item scale measuring the sense of social and academic fit of students at the institution where this study was conducted. Higher values indicate higher feelings of belongings. | 17 - 119 | | |
| 2-Way SSS [80] 2-Way Social Support Scale | A 21-item scale measuring social supports from four aspects (a) giving emotional support, (b) giving instrumental support, (c) receiving emotional support, and (d) receiving instrumental support. Higher values indicate more social support. | (a) 0 - 25 (b) 0 - 25 (c) 0 - 35 (d) 0 - 20 | | |
| PSS [24] Perceived Stress Scale | A 14-item scale used to assess stress levels during the last month. Note that Year 1 used the 10-item version. Higher values indicate more perceived stress. | 0 - 56 (Year 2,3,4) 0 - 40 (Year 1) | | |
| ERQ [45] Emotion Regulation Questionnaire | A 10-item scale assessing individual differences in the habitual use of two emotion regulation strategies: (a) cognitive reappraisal and (b) expressive suppression. Higher scores indicate more habitual use of reappraisal/suppression. | (a) 1 - 7 (b) 1 - 7 | | |
| BRS [81] Brief Resilience Scale | A 6-item scale assessing the ability to bounce back or recover from stress. Higher scores indicate more resilient from stress. | 1 - 5 | | |
| CHIPS [23] Cohen-Hoberman Inventory of Physical Symptoms | A 33-item scale measuring the perceived burden from physical symptoms, and resulting psychological effect during the past 2 weeks. Higher values indicate more perceived burden from physical symptoms. | 0 - 132 | | |
| STAI [13, 47] State-Trait Anxiety Inventory for Adults | A 20-item scale measuring State-Trait anxiety. Year 1 used the State version, while other years used the Trait version. Higher values indicate higher anxiety. | 20 - 80 | | |
| CES-D [25, 74] Center for Epidemiologic Studies Depression Scale Cole version | A 10-item scale measuring current level of depressive symptomatology, with emphasis on the affective component, depressed mood. Year 2 used the 9-item version. Higher scores indicate more depressive symptoms. | 0 - 30 (Year 1,3,4) 0 - 27 (Year 2) | | |
| BDI2 [11] Beck Depression Inventory-II | A 21-item detect depressive symptoms. Higher values indicate more depressive symptoms. 0-13: minimal to none, 14-19: mild, 20-28: moderate and 26-63: severe. | 0 - 63 | | |
| MAAS [16] Mindful Attention Awareness Scale | A 15-item scale assessing a core characteristic of mindfulness. Year 1 used a 7-item version, while other years used the full version. Higher values indicate higher mindfulness. | 1 - 6 | | |
| BFI10 [75] The Big-Five Inventory-10 | A 10-item scale measuring the Big Five personality traits Extroversion, Agreeableness, Conscientiousness, Emotional Stability, and Openness. The higher the score, the greater the tendency of the corresponding personality. | 1 - 5 | 1,2,3,4 | pre |
| Brief-COPE [19] Brief Coping Orientation to Problems Experienced | A 28-item scale measuring (a) adaptive and (b) maladaptive ways to cope with a stressful life event. Higher values indicate more effective/ineffective ways to cope with a stressful life event. | (a): 0 - 3 (b): 0 - 3 | 2,3,4 | pre, post |
| GQ [64] Gratitude Questionnaire | A 6-item scale assessing individual differences in the proneness to experience gratitude in daily life. Higher scores indicate a greater tendency to experience gratitude. | 6 - 42 | | |
| FSPWB [28] Flourishing Scale & Psychological Well-Being Scale | An 8-item scale measuring the psychological well-being. Higher scores indicate a person with "more psychological resources and mental strengths". | 8 - 56 | | |
| EDS [5, 100] Everyday Discrimination Scale | A 9-item scale assessing everyday discrimination. Higher values indicate more frequent experience of discrimination. | 0 - 45 | | |
| CEDH [14, 100] Chronic Work Discrimination and Harassment | A 12-item scale assessing experiences of discrimination in educational settings. Higher values indicate more frequent experience of discrimination in the work environment. | 0 - 60 | | |
| B-YAACQ [48] The Brief Young Adult Alcohol Consequences Questionnaire (optional) | A 24-item scale measuring the alcohol problem severity continuum in college students. Higher values indicates more severe alcohol problems. | 0 - 24 | | |
| PHQ-4 [6, 51] Patient Health Questionnaire 4 | A 4-item scale assessing (a) mental health, (b) anxiety, and (c) depression. Higher values indicate higher risk of mental health, anxiety, and depression. | (a): 0 - 12 (b): 0 - 6 (c): 0 - 6 | 2,3,4 | Weekly EMA |
| PSS-4 [1, 24] Perceived Stress Scale 4 | A 4-item scale assessing stress levels during the last month. Higher values indicates more perceived stress. | 0 - 16 | | |
| PANAS [2, 99] Positive and Negative Affect Schedule | A 10-item scale measuring the level of (a) positive and (b) negative affects. Higher values indicates larger extent. | (a): 0 - 20 (b): 0 - 20 | | |

## A.7 Sensor Feature Details

The following tables list out specific features based on RAPIDS [88]. All features are extracted with multiple `time_segments`: morning (6 am - 12 pm), afternoon (12 pm - 6 pm), evening (6 pm - 12 am), night (12 am - 6 am), allday, 7-day history, 14-day history, weekday, and weekend (the last two are calculated once a week). Moreover, all numeric features have two extra versions: 1) normalized (subtracted by each participant's median and divided by the 5-95 quantile range); 2) discretized (low/medium/high split by 33/66 quantile of each participant's feature value). We employ a specific naming format of all features:

$$[\texttt{feature\_type}]:[\texttt{feature\_name}][\texttt{\_norm or NULL}]:[\texttt{time\_segment}]$$

Table 5: Description of Location Features. Texts taken from RAPIDS with courtesy. "Missing" column indicate the missing rate of the corresponding feature(s). The same below.

| Feature Type | Feature Name | Unit | Missing | Description |
|---|---|---|---|---|
| **Location** | hometime | minutes | 23.2% | Time at home. Time spent at home in minutes. Home is the most visited significant location between 8 pm and 8 am, including any pauses within a 200-meter radius. |
| | disttravelled | meters | 23.2% | Total distance traveled over a day (flights). |
| | rog | meters | 23.2% | The Radius of Gyration (rog) is a measure in meters of the area covered by a person over a day. A centroid is calculated for all the places (pauses) visited during a day, and a weighted distance between all the places and that centroid is computed. The weights are proportional to the time spent in each place. |
| | maxdiam | meters | 23.2% | The maximum diameter is the largest distance between any two pauses. |
| | maxhomedist | meters | 23.2% | The maximum distance from home in meters. |
| | siglocsvisited | locations | 23.2% | The number of significant locations visited during the day. Significant locations are computed using k-means clustering over pauses found in the whole monitoring period. The number of clusters is found iterating k from 1 to 200 stopping until the centroids of two significant locations are within 400 meters of one another. |
| | avgflightlen | meters | 23.2% | Mean length of all flights. |
| | stdflightlen | meters | 23.2% | Standard deviation of the length of all flights. |
| | avgflightdur | seconds | 23.2% | Mean duration of all flights. |
| | stdflightdur | seconds | 23.2% | The standard deviation of the duration of all flights. |
| | probpause | - | 23.2% | The fraction of a day spent in a pause (as opposed to a flight). |
| | siglocentropy | nats | 23.2% | Shannon's entropy measurement is based on the proportion of time spent at each significant location visited during a day. |
| | circdnrtn | - | 23.2% | A continuous metric quantifying a person's circadian routine that can take any value between 0 and 1, where 0 represents a daily routine completely different from any other sensed days and 1 a routine the same as every other sensed day. |
| | wkenddayrtn | - | 23.2% | Same as circdnrtn but computed separately for weekends and weekdays. |
| | locationvariance | meters2 | 14.5% | The sum of the variances of the latitude and longitude columns. |
| | loglocationvariance | - | 14.7% | Log of the sum of the variances of the latitude and longitude columns. |
| | totaldistance | meters | 14.5% | Total distance traveled in a time segment using the haversine formula. |
| | avgspeed | km/hr | 14.5% | Average speed in a time segment considering only the instances labeled as Moving. This feature is 0 when the participant is stationary during a time segment. |
| | varspeed | km/hr | 14.5% | Speed variance in a time segment considering only the instances labeled as Moving. This feature is 0 when the participant is stationary during a time segment. |
| | numberofsignificantplaces | places | 14.5% | Number of significant locations visited. It is calculated using the DBSCAN/OPTICS clustering algorithm which takes in EPS and MIN_SAMPLES as parameters to identify clusters. Each cluster is a significant place. |
| | numberlocationtransitions | transi-tions | 14.5% | Number of movements between any two clusters in a time segment. |
| | radiusgyration | meters | 14.5% | Quantifies the area covered by a participant. |
| | timeattop1location | minutes | 14.5% | Time spent at the most significant location. |
| | timeattop2location | minutes | 14.5% | Time spent at the 2nd most significant location. |
| | timeattop3location | minutes | 14.5% | Time spent at the 3rd most significant location. |
| | movingtostaticratio | - | 14.5% | Ratio between stationary time and total location sensed time. A lat/long coordinate pair is labeled as stationary if its speed (distance/time) to the next coordinate pair is less than 1km/hr. A higher value represents a more stationary routine. |
| | outlierstimepercent | - | 14.5% | Ratio between the time spent in non-significant clusters divided by the time spent in all clusters (stationary time. Only stationary samples are clustered). A higher value represents more time spent in non-significant clusters. |
| | maxlengthstayatclusters | minutes | 14.5% | Maximum time spent in a cluster (significant location). |
| | minlengthstayatclusters | minutes | 14.5% | Minimum time spent in a cluster (significant location). |
| | avglengthstayatclusters | minutes | 14.5% | Average time spent in a cluster (significant location). |
| | stdlengthstayatclusters | minutes | 14.5% | Standard deviation of time spent in a cluster (significant location). |
| | locationentropy | nats | 14.5% | Shannon Entropy computed over the row count of each cluster (significant location), it is higher the more rows belong to a cluster (i.e., the more time a participant spent at a significant location). |
| | normalizedlocationentropy | nats | 14.5% | Shannon Entropy computed over the row count of each cluster (significant location) divided by the number of clusters; it is higher the more rows belong to a cluster (i.e., the more time a participant spent at a significant location). |
| | timeathome | minutes | 14.5% | Time spent at home. |
| | timeat [PLACE] | minutes | 14.5% | Time spent at [PLACE], which can be living, exercise, study, greens. |

Table 6: Description of Phone Usage, Call, and Bluetooth Features

| Feature Type | Feature Name | Unit | Missing | Description |
|---|---|---|---|---|
| **Phone Usage** | sumduration | minutes | 14.4% | Total duration of all unlock episodes. |
| | maxduration | minutes | 14.4% | Longest duration of any unlock episode. |
| | minduration | minutes | 14.4% | Shortest duration of any unlock episode. |
| | avgduration | minutes | 14.4% | Average duration of all unlock episodes. |
| | stdduration | minutes | 14.8% | Standard deviation duration of all unlock episodes. |
| | countepisode | episodes | 14.4% | Number of all unlock episodes. |
| | firstuseafter | minutes | 14.4% | Minutes until the first unlock episode. |
| | sumduration [PLACE] | minutes | 14.4% | Total duration of all unlock episodes. [PLACE] can be living, exercise, study, greens. Same below. |
| | maxduration [PLACE] | minutes | 14.4% | Longest duration of any unlock episode. |
| | minduration [PLACE] | minutes | 14.4% | Shortest duration of any unlock episode. |
| | avgduration [PLACE] | minutes | 14.4% | Average duration of all unlock episodes. |
| | stdduration [PLACE] | minutes | 14.8% | Standard deviation duration of all unlock episodes. |
| | countepisode [PLACE] | episodes | 14.4% | Number of all unlock episodes. |
| | firstuseafter [PLACE] | minutes | 14.4% | Minutes until the first unlock episode. |
| **Call** | count | calls | 51.6% | Number of calls of a particular call_type (either incoming or outgoing, same below) occurred during a particular time_segment. |
| | distinctcontacts | contacts | 51.6% | Number of distinct contacts that are associated with a particular call_type for a particular time_segment. |
| | meanduration | seconds | 63.6% | The mean duration of all calls of a particular call_type during a particular time_segment. |
| | sumduration | seconds | 63.6% | The sum of the duration of all calls of a particular call_type during a particular time_segment. |
| | minduration | seconds | 63.6% | The duration of the shortest call of a particular call_type during a particular time_segment. |
| | maxduration | seconds | 63.6% | The duration of the longest call of a particular call_type during a particular time_segment. |
| | stdduration | seconds | 76.2% | The standard deviation of the duration of all the calls of a particular call_type during a particular time_segment. |
| | modeduration | seconds | 63.6% | The mode of the duration of all the calls of a particular call_type during a particular time_segment. |
| | entropyduration | nats | 65.9% | The estimate of the Shannon entropy for the the duration of all the calls of a particular call_type during a particular time_segment. |
| | timefirstcall | minutes | 63.6% | The time in minutes between 12:00am (midnight) and the first call of call_type. |
| | timelastcall | minutes | 63.6% | The time in minutes between 12:00am (midnight) and the last call of call_type. |
| | countmostfrequentcontact | calls | 51.6% | The number of calls of a particular call_type during a particular time_segment of the most frequent contact throughout the monitored period. |
| **Bluetooth** | countscans | scans | 23.7% | Number of scans (rows) from the devices sensed during a time segment instance. The more scans a bluetooth device has the longer it remained within range of the participant's phone. |
| | uniquedevices | devices | 23.7% | Number of unique bluetooth devices sensed during a time segment instance as identified by their hardware addresses (bt_address). |
| | meanscans | scans | 23.7% | Mean of the scans of every sensed device within each time segment instance. |
| | stdscans | scans | 35.1% | Standard deviation of the scans of every sensed device within each time segment instance. |
| | countscansmostfrequent devicewithinsegments | scans | 23.7% | Number of scans of the most sensed device within each time segment instance. |
| | countscansleastfrequent devicewithinsegments | scans | 23.7% | Number of scans of the least sensed device within each time segment instance. |
| | countscansmostfrequent deviceacrosssegments | scans | 23.7% | Number of scans of the most sensed device across time segment instances of the same type. |
| | countscansleastfrequent deviceacrosssegments | scans | 23.7% | Number of scans of the least sensed device across time segment instances of the same type per device. |
| | countscansmostfrequent deviceacrossdataset | scans | 23.7% | Number of scans of the most sensed device across the entire dataset of every participant. |
| | countscansleastfrequent deviceacrossdataset | scans | 23.7% | Number of scans of the least sensed device across the entire dataset of every participant. |

Table 7: Description of Physical Activity and Sleep Features

| Feature Type | Feature Name | Unit | Missing | Description |
|---|---|---|---|---|
| **Physical Activity** | maxsumsteps | steps | 29.2% | The maximum daily step count during a time segment. |
| | minsumsteps | steps | 29.2% | The minimum daily step count during a time segment. |
| | avgsumsteps | steps | 29.2% | The average daily step count during a time segment. |
| | mediansumsteps | steps | 29.2% | The median of daily step count during a time segment. |
| | stdsumsteps | steps | 29.2% | The standard deviation of daily step count during a time segment. |
| | sumsteps | steps | 29.3% | The total step count during a time segment. |
| | maxsteps | steps | 29.3% | The maximum step count during a time segment. |
| | minsteps | steps | 29.3% | The minimum step count during a time segment. |
| | avgsteps | steps | 29.3% | The average step count during a time segment. |
| | countepisodesedentarybout | bouts | 29.3% | Number of sedentary bouts during a time segment. |
| | sumdurationsedentarybout | minutes | 29.3% | Total duration of all sedentary bouts during a time segment. |
| | maxdurationsedentarybout | minutes | 29.3% | The maximum duration of any sedentary bout during a time segment. |
| | mindurationsedentarybout | minutes | 29.3% | The minimum duration of any sedentary bout during a time segment. |
| | avgdurationsedentarybout | minutes | 29.3% | The average duration of sedentary bouts during a time segment. |
| | stddurationsedentarybout | minutes | 29.3% | The standard deviation of the duration of sedentary bouts during a time segment. |
| | countepisodeactivebout | bouts | 29.3% | Number of active bouts during a time segment. |
| | sumdurationactivebout | minutes | 29.3% | Total duration of all active bouts during a time segment. |
| | maxdurationactivebout | minutes | 29.3% | The maximum duration of any active bout during a time segment. |
| | mindurationactivebout | minutes | 29.3% | The minimum duration of any active bout during a time segment. |
| | avgdurationactivebout | minutes | 29.3% | The average duration of active bouts during a time segment. |
| | stddurationactivebout | minutes | 29.3% | The standard deviation of the duration of active bouts during a time segment. |
| **Sleep** | countepisode [LEVEL][TYPE] | episodes | 34.5% | Number of [LEVEL][TYPE] sleep episodes. [LEVEL] is one of awake and asleep and [TYPE] is one of main, nap, and all. Same below. |
| | sumduration [LEVEL][TYPE] | minutes | 34.5% | Total duration of all [LEVEL][TYPE] sleep episodes. |
| | maxduration [LEVEL][TYPE] | minutes | 34.5% | Longest duration of any [LEVEL][TYPE] sleep episode. |
| | minduration [LEVEL][TYPE] | minutes | 34.5% | Shortest duration of any [LEVEL][TYPE] sleep episode. |
| | avgduration [LEVEL][TYPE] | minutes | 34.5% | Average duration of all [LEVEL][TYPE] sleep episodes. |
| | medianduration [LEVEL][TYPE] | minutes | 34.5% | Median duration of all [LEVEL][TYPE] sleep episodes. |
| | stdduration [LEVEL][TYPE] | minutes | 34.5% | Standard deviation duration of all [LEVEL][TYPE] sleep episodes. |
| | firstwaketime [TYPE] | minutes | 36.4% | First wake time for a certain sleep type during a time segment. Wake time is number of minutes after midnight of a sleep episode's end time. |
| | lastwaketime [TYPE] | minutes | 36.4% | Last wake time for a certain sleep type during a time segment. Wake time is number of minutes after midnight of a sleep episode's end time. |
| | firstbedtime [TYPE] | minutes | 36.3% | First bedtime for a certain sleep type during a time segment. Bedtime is number of minutes after midnight of a sleep episode's start time. |
| | lastbedtime [TYPE] | minutes | 36.3% | Last bedtime for a certain sleep type during a time segment. Bedtime is number of minutes after midnight of a sleep episode's start time. |
| | countepisode [TYPE] | episodes | 34.5% | Number of sleep episodes for a certain sleep type during a time segment. |
| | avgefficiency [TYPE] | scores | 36.3% | Average sleep efficiency for a certain sleep type during a time segment. |
| | sumdurationafterwakeup [TYPE] | minutes | 35.6% | Total duration the user stayed in bed after waking up for a certain sleep type during a time segment. |
| | sumdurationasleep [TYPE] | minutes | 34.5% | Total sleep duration for a certain sleep type during a time segment. |
| | sumdurationawake [TYPE] | minutes | 34.5% | Total duration the user stayed awake but still in bed for a certain sleep type during a time segment. |
| | sumdurationtofallasleep [TYPE] | minutes | 35.6% | Total duration the user spent to fall asleep for a certain sleep type during a time segment. |
| | sumdurationinbed [TYPE] | minutes | 35.6% | Total duration the user stayed in bed (sumdurationtofallasleep + sumdurationawake + sumdurationasleep + sumdurationafterwakeup) for a certain sleep type during a time segment. |
| | avgdurationafterwakeup [TYPE] | minutes | 35.6% | Average duration the user stayed in bed after waking up for a certain sleep type during a time segment. |
| | avgdurationasleep [TYPE] | minutes | 34.5% | Average sleep duration for a certain sleep type during a time segment. |
| | avgdurationawake [TYPE] | minutes | 34.5% | Average duration the user stayed awake but still in bed for a certain sleep type during a time segment. |
| | avgdurationtofallasleep [TYPE] | minutes | 35.6% | Average duration the user spent to fall asleep for a certain sleep type during a time segment. |
| | avgdurationinbed [TYPE] | minutes | 35.6% | Average duration the user stayed in bed (sumdurationtofallasleep + sumdurationawake + sumdurationasleep + sumdurationafterwakeup) for a certain sleep type during a time segment. |

PS.1. It is worth noting that the missing rate of call-related features are high. This is mainly because most these features are event-based. If a participant did not receive a phone call at a day, that day will have empty call features.

PS.2. One limitation of our physical activity and sleep feature data comes from a Fitbit issue: If the data on the wearable device is not synced with the smartphone over a few days, it would trigger some internal space-saving strategy to discard low-level details and only contain high-level summary data, leading to information loss and affecting feature correctness. This would be reflected by the missing features (*e.g.*, small or missing `countepisodeactivebout`), which is not common in our datasets.

# B  Additional Model & Benchmark Information

We provide a more detailed description of benchmark-related processing. Many texts are taken from [104] with courtesy.

## B.1  Depression Detection Ground Truth Processing

Due to some design iteration, we did not include PHQ-4 in DS1, but only PANAS. Although PANAS contains questions related to depressive symptoms (*e.g.*, "distressed"), it does not have a comparable theoretical foundation for depression detection like PHQ-4 or BDI-II. Therefore, to maximize the compatibility of the datasets, we trained a small ML model on DS2 that has both PANAS and PHQ-4 scores to generate reliable ground truth labels. Specifically, we used a decision tree (depth=2) to take PNANS scores on two affect questions ("depressed" and "nervous") as the input and predict PHQ-4 score-based depression binary label. Our model achieved 74.5% and 76.3% for accuracy and F1-score on a 5-fold cross-validation on DS2. The rule from the decision tree is simple: the user would be labeled as having no depression when the distress score is less than 2, and the nervous score is less than 3 (on a 1-5 Likert Scale). We then applied this rule to DS1 to generate depression labels.

## B.2  Behavior Modeling Algorithm Implementation Details

Please refer to our GLOBEM codebease for the specific implementations and hyperparameter tuning.

### B.2.1  Depression Detection Algorithms

1. *Canzian et al.* [18]
   Features: Location trajectory features directly computed from the past two-week time window.
   Model: A support vector machine (SVM).
2. *Saeb et al.* [78]
   Features: Location and screen features aggregated with daily average of the past two weeks.
   Model: A logistic regression model with elastic regularization.
3. *Farhan et al.* [33]
   Features: Location and physical activity features from the past two-week window.
   Model: An SVM.
4. *Wahle et al.* [91]
   Features: Several feature types (activity, location, WiFi, screen, and call) over the past two weeks. Both daily aggregation (*i.e.*, mean, sum, variance) and direct computation of the features of the two weeks are used. WiFi features are excluded to ensure the compatibility with our datasets.
   Model: SVM and Random Forest.
5. *Lu et al.* [60]
   Features: Location, activity, and sleep features computed from the past two weeks.
   Model: Multi-task learning combining linear regression & logistic regression. One model for iOS and one for Android are built to deal with device platform differences,
6. *Wang et al.* [97]
   Features: Location, screen, activity, sleep, and audio features aggregated by calculating daily average and slope of the past two weeks. Audio features are excluded as they are not collected.
   Model: A lasso-regularized logistic regression model.
7. *Xu et al.-I (Interpretable)* [101]
   Features: Location, screen, activity, and sleep features in multiple epochs of a day (morning, afternoon, evening, night). Association rule mining is applied to mine out interpretable behavior rules that capture differences between participants with depression and without depression. Then, the rules are used to filter and aggregate features of multiple days.
   Model: An Adaboost model.
8. *Xu et al.-P (Personalized)* [102]
   Features: A similar set of basic features as [101]. With each feature as a time sequence, a user behavior relevance matrix is computed using the square of Pearson correlation to capture users with strong positive or negative correlation.
   Model: aAtraditional collaborative-filtering-based model to select features and obtain an intermediate prediction using each feature, and combine the results of all features via majority voting.

9. *Chikersal et al.* [20]

   Features: A similar set of basic features as [101]. Aggregations (breakpoint and slope) across multiple time ranges (daily and biweekly) are calculated, followed by a nested randomized logistic regression for feature selection.

   Model: Separate gradient boosting and logistic regression models using data from every sensor, and combine the prediction with another Adaboost model to generate the final prediction.

Each algorithm will lead to one model. All these models' hyperparameters are tuned via grid search with the same range as mentioned in each prior work.

### B.2.2   Domain Generalization Algorithms

The data format of all deep-learning based algorithm is the same: a subset of important daily features in the most recent traditional depression detection algorithms [20, 102], with the past-four-week feature matrix as the input. It is worth noting that we picked these deep learning techniques to cover the major approaches of domain generalization [94], including 1) data manipulation (Mixup), 2) representation learning (IRM, DANN, CSD), and 3) learning strategy (MLDG, MASF, Siamese, Reorder).

1. *ERM* (Empirical Risk Minimization) [87]
   The basic model training techniques without particular design for domain generalization. ERM shows a competitive performance in previous CV generalization tasks [46, 94]. Multiple architectures with ERM are implemented: a) *ERM-1D-CNN*: one-dimensional CNN that treats the data as a time-series of length 28; b) *ERM-2D-CNN*: two-dimensional CNN that treats the data as an one-channel image; c) *ERM-LSTM*: another architecture to model time-series data; d) *ERM-Transformer*: a transformer-based architecture for modeling sequence data.

2. *Mixup* (ERM-Mixup) [111]
   A data augmentation technique that performs linear interpolation between two instances with a weight sampled from a Beta distribution. 1D-CNN is used as the architecture as it is robust to feature positions in the feature matrix. Same for the rest algorithms.

3. *IRM* (Invariant Risk Minimization) [7]
   A representation learning paradigm to estimate invariant correlations across multiple distributions and learn a data representation such that the optimal classifier can match all training distributions.

4. *DANN* (Domain-Adversarial Neural Network) [38]
   Another representation learning technique that adversarially trains the generator and discriminator. The discriminator is trained to distinguish different domains, while the generator is trained to fool the discriminator to learn domain-invariant feature representations. Two setups are tested, one treating each dataset as a domain (*DANN-D (Dataset as Domain)*), and one treating each person as a domain (*DANN-P (Person as Domain)*).

5. *CSD* (Common Specific Decomposition) [72]
   A feature disentanglement-based representation learning technique from the multi-component analysis perspective, which extracts the domain-shared and domain-specific features using separate network parameters. Similar to DANN, it can support *CSD-D* and *CSD-P*.

6. *MLDG* (Meta-Learning for Domain Generalization) [56]
   One of the first methods using meta-learning strategy for domain generalization. MLDG splits the data of the training domains into meta-train and meta-test to simulate the domain shift to learn general features. It supports *MLDG-D*, and *MLDG-P*.

7. *MASF* (Model-Agnostic Learning of Semantic Features) [30]
   A learning strategy that combines meta-learning and feature disentanglement. After simulating domain shift by domain split, MASF further regularizes the semantic structure of the feature space by introducing a global loss (to preserve relationships between classes) and a local loss (to promote domain-independent class clustering). It supports *MASF-D*, and *MASF-P*.

8. *Siamese Network* [49]
   A metric-learning based strategy to find a better pair-wise distance metric. It aims to decrease the distance between positive pairs and increase the distance between negative pairs.

9. *Reorder* [104]
   A recently proposed method to leverage the continuity of behavior trajectory [104]. It designed a pretext task which shuffles the temporal order of the feature matrix. Then a model is trained to reconstruct the original sequence, jointly optimized with the main classification task over different

domains, as shown in Fig.9. By capturing the continuity of daily behaviors, the model could learn to extract representations that are generalizable across individuals. Overall, the model can be trained via the following objective function:

$$\operatorname*{argmin}_{\theta_f,\theta_c,\theta_r} \sum_{i=1}^{S} \left( \sum_{j=1}^{N_i} \mathcal{L}_c(h(x_j^i|\theta_f,\theta_c),y_j^i) + \sum_{j=1}^{\beta N_i} \alpha \mathcal{L}_r(h(z_j^i|\theta_f,\theta_r),p_j^i) \right)$$

where both $\mathcal{L}_c$ and $\mathcal{L}_r$ are cross-entropy losses. $S$ is the total number of training domains, and $N_i$ is the size of a domain $i$. $\alpha$ is used to control the weight of the reordering task while $\beta$ is used to control the size of reordering data. $x$ is the input matrix, $y$ is the classification label, $z$ is the feature matrix $x$ after the reordering, and $p$ is the permutation index (from 1 to 200 among the 200 pre-determined permutation set). $x_j^i, y_j^i, z_j^i, p_j^i$ are specific instances in each domain $i$ with index $j$. We picked the number of segmentation as $n = 10$ ($\lceil 28/3 \rceil$) since 28! or 14! (28/2) is too computationally expensive.

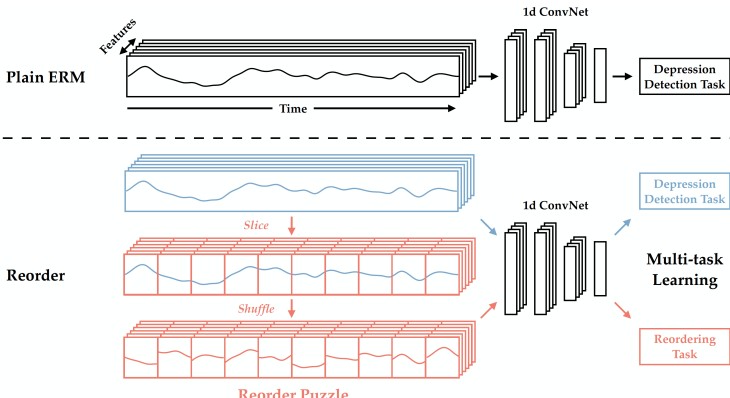

Figure 9: The Design of Reorder Compared to ERM (taken from [104] with courtesy).

One algorithm could lead to one or multiple models. Models from No.2 to No.9 all use the same 1D-CNN as the backbone. We use a simple architecture based on a small-range tuning using ERM-1D-CNN. It has 3 1D-convolution layers (size 8, stride 3, ReLU activation), each followed by a batch normalization layer, a max-pooling layer, as well as a dropout layer (rate 0.25). We tested with different layer sizes (8, 16, 32) and depth (3,5,7), and observed similar results, thus we chose size as 8 and depth as 3 to save computing cost. A fully connected layer (size 16) was attached after flattening the third convolution layer's output to convert it into a vector of length 16. The following layers are customized for each model.

Other architectures are also simple: *ERM-2D-CNN* used three 2D-convolution layers with the same size, stride, and activation function as 1D-CNN; *ERM-LSTM* used two bi-directional layers with the hidden size as 20; *ERM-Transformer* used two transformer blocks, each with 4 self-attention heads (size 4) and a 1D-convolutional feed forward layer (size 16).

For all models, we adopted a common training setup. Specifically, we used Adam as the optimizer and adopted a cosine annealing schedule, with an initial learning rate of 0.001, an annealing decay of 0.95, and a step size of 100.

### B.2.3   Training Resources

Since all deep learning models are small, we only used CPU for the model training. We leveraged a university computing cluster (300 CPUs) with the SLURM Workload Manager. The whole training was completed within 48 hours.

### B.3   Additional Generalization Results

Table 8: Model Performance of Depression Detection in Single Dataset.

| Category | Model | Balanced Accuracy | | | | | ROC AUC | | | | |
|---|---|---|---|---|---|---|---|---|---|---|---|
| | | DS1 | DS2 | DS3 | DS4 | Avg | DS1 | DS2 | DS3 | DS4 | Avg |
| Baseline | Majority | 0.500 | 0.500 | 0.500 | 0.500 | 0.500 | 0.500 | 0.500 | 0.500 | 0.500 | 0.500 |
| Prior Depression Detection Model | Canzian et al. [18] | 0.500 | 0.500 | 0.608 | 0.536 | 0.536 | 0.597 | 0.514 | 0.626 | 0.607 | 0.586 |
| | Saeb et al. [78] | 0.526 | 0.533 | 0.613 | 0.557 | 0.557 | 0.555 | 0.581 | 0.641 | 0.614 | 0.598 |
| | Farhan et al. [33] | 0.554 | 0.509 | 0.604 | 0.582 | 0.562 | 0.575 | 0.554 | 0.665 | 0.618 | 0.603 |
| | Wahle et al. [91] | 0.584 | 0.548 | 0.632 | 0.628 | 0.598 | 0.611 | 0.568 | 0.665 | 0.702 | 0.637 |
| | Lu et al. [60] | 0.529 | 0.496 | 0.604 | 0.569 | 0.550 | 0.530 | 0.499 | 0.674 | 0.599 | 0.576 |
| | Wang et al. [97] | 0.548 | 0.500 | 0.494 | 0.578 | 0.530 | 0.610 | 0.500 | 0.491 | 0.653 | 0.564 |
| | Xu et al.-I [101] | 0.669 | 0.655 | 0.731 | 0.710 | 0.691 | 0.699 | 0.706 | 0.759 | 0.786 | 0.737 |
| | Xu et al.-P [102] | 0.591 | 0.612 | 0.611 | 0.584 | 0.600 | 0.632 | 0.637 | 0.621 | 0.632 | 0.630 |
| | Chikersal et al. [20] | 0.656 | 0.611 | 0.641 | 0.690 | 0.649 | 0.726 | 0.679 | 0.695 | 0.763 | 0.716 |
| Recent Domain Generalization Model | ERM-1dCNN [87] | 0.579 | 0.556 | 0.578 | 0.560 | 0.568 | 0.608 | 0.558 | 0.599 | 0.618 | 0.596 |
| | ERM-2dCNN [87] | 0.506 | 0.535 | 0.524 | 0.567 | 0.533 | 0.541 | 0.530 | 0.530 | 0.575 | 0.544 |
| | ERM-LSTM [87] | 0.579 | 0.554 | 0.519 | 0.607 | 0.565 | 0.583 | 0.573 | 0.529 | 0.630 | 0.579 |
| | ERM-Transformer [87] | 0.574 | 0.619 | 0.556 | 0.586 | 0.584 | 0.604 | 0.636 | 0.557 | 0.612 | 0.602 |
| | ERM-Mixup [111] | 0.579 | 0.556 | 0.578 | 0.560 | 0.568 | 0.608 | 0.558 | 0.599 | 0.618 | 0.596 |
| | IRM [7] | 0.571 | 0.529 | 0.595 | 0.599 | 0.573 | 0.607 | 0.568 | 0.642 | 0.650 | 0.617 |
| | DANN-D [39] | 0.564 | 0.511 | 0.489 | 0.538 | 0.526 | 0.557 | 0.502 | 0.487 | 0.575 | 0.530 |
| | DANN-P [39] | 0.508 | 0.500 | 0.500 | 0.500 | 0.502 | 0.523 | 0.490 | 0.563 | 0.552 | 0.532 |
| | CSD-D [72] | 0.591 | 0.502 | 0.596 | 0.557 | 0.562 | 0.601 | 0.536 | 0.612 | 0.631 | 0.595 |
| | CSD-P [72] | 0.550 | 0.513 | 0.544 | 0.559 | 0.542 | 0.581 | 0.505 | 0.568 | 0.613 | 0.567 |
| | MLDG-D [56] | 0.550 | 0.539 | 0.495 | 0.504 | 0.522 | 0.573 | 0.515 | 0.520 | 0.507 | 0.529 |
| | MLDG-P [56] | 0.529 | 0.517 | 0.478 | 0.507 | 0.508 | 0.554 | 0.499 | 0.473 | 0.523 | 0.512 |
| | MASF-D [30] | 0.489 | 0.518 | 0.505 | 0.506 | 0.505 | 0.509 | 0.531 | 0.492 | 0.541 | 0.518 |
| | MASF-P [30] | 0.486 | 0.492 | 0.487 | 0.515 | 0.495 | 0.503 | 0.502 | 0.501 | 0.514 | 0.505 |
| | Siamese Network [49] | 0.570 | 0.481 | 0.533 | 0.596 | 0.545 | 0.570 | 0.481 | 0.533 | 0.596 | 0.545 |
| | Reorder [104] | 0.616 | 0.606 | 0.639 | 0.644 | 0.626 | 0.657 | 0.619 | 0.671 | 0.692 | 0.660 |

Table 9: Model Performance of Depression Detection with Leave-One-Dataset-Out Setup.

| Category | Model | Balanced Accuracy | | | | | ROC AUC | | | | |
|---|---|---|---|---|---|---|---|---|---|---|---|
| | | DS1 | DS2 | DS3 | DS4 | Avg | DS1 | DS2 | DS3 | DS4 | Avg |
| Baseline | Majority | 0.500 | 0.500 | 0.500 | 0.500 | 0.500 | 0.500 | 0.500 | 0.500 | 0.500 | 0.500 |
| Prior Depression Detection Model | Canzian et al. [18] | 0.480 | 0.504 | 0.506 | 0.501 | 0.498 | 0.491 | 0.484 | 0.480 | 0.542 | 0.499 |
| | Saeb et al. [78] | 0.525 | 0.536 | 0.523 | 0.558 | 0.536 | 0.529 | 0.548 | 0.529 | 0.567 | 0.543 |
| | Farhan et al. [33] | 0.505 | 0.497 | 0.496 | 0.525 | 0.506 | 0.505 | 0.550 | 0.515 | 0.553 | 0.531 |
| | Wahle et al. [91] | 0.526 | 0.527 | 0.495 | 0.546 | 0.524 | 0.543 | 0.554 | 0.503 | 0.564 | 0.541 |
| | Lu et al. [60] | 0.546 | 0.498 | 0.541 | 0.538 | 0.531 | 0.550 | 0.510 | 0.588 | 0.564 | 0.553 |
| | Wang et al. [97] | 0.509 | 0.521 | 0.515 | 0.541 | 0.521 | 0.514 | 0.556 | 0.529 | 0.554 | 0.538 |
| | Xu et al.-I [101] | 0.517 | 0.525 | 0.474 | 0.494 | 0.502 | 0.512 | 0.527 | 0.477 | 0.484 | 0.500 |
| | Xu et al.-P [102] | 0.508 | 0.501 | 0.486 | 0.512 | 0.502 | 0.545 | 0.535 | 0.504 | 0.521 | 0.526 |
| | Chikersal et al. [20] | 0.540 | 0.534 | 0.531 | 0.538 | 0.536 | 0.555 | 0.561 | 0.558 | 0.545 | 0.555 |
| Recent Domain Generalization Model | ERM-1dCNN [87] | 0.490 | 0.527 | 0.508 | 0.514 | 0.510 | 0.487 | 0.532 | 0.490 | 0.524 | 0.508 |
| | ERM-2dCNN [87] | 0.511 | 0.495 | 0.507 | 0.525 | 0.510 | 0.514 | 0.499 | 0.509 | 0.534 | 0.514 |
| | ERM-LSTM [87] | 0.514 | 0.519 | 0.494 | 0.522 | 0.512 | 0.521 | 0.525 | 0.480 | 0.528 | 0.514 |
| | ERM-Transformer [87] | 0.492 | 0.506 | 0.531 | 0.507 | 0.509 | 0.499 | 0.513 | 0.526 | 0.510 | 0.512 |
| | ERM-Mixup [111] | 0.498 | 0.524 | 0.493 | 0.489 | 0.501 | 0.506 | 0.538 | 0.498 | 0.495 | 0.509 |
| | IRM [7] | 0.492 | 0.519 | 0.511 | 0.503 | 0.506 | 0.500 | 0.533 | 0.521 | 0.517 | 0.518 |
| | DANN-D [39] | 0.509 | 0.508 | 0.514 | 0.527 | 0.514 | 0.511 | 0.505 | 0.516 | 0.536 | 0.517 |
| | DANN-P [39] | 0.500 | 0.500 | 0.500 | 0.500 | 0.500 | 0.502 | 0.484 | 0.485 | 0.518 | 0.497 |
| | CSD-D [72] | 0.521 | 0.521 | 0.515 | 0.527 | 0.521 | 0.525 | 0.526 | 0.525 | 0.539 | 0.529 |
| | CSD-P [72] | 0.500 | 0.513 | 0.506 | 0.526 | 0.511 | 0.499 | 0.520 | 0.507 | 0.541 | 0.517 |
| | MLDG-D [56] | 0.513 | 0.526 | 0.508 | 0.495 | 0.511 | 0.525 | 0.536 | 0.505 | 0.495 | 0.515 |
| | MLDG-P [56] | 0.509 | 0.503 | 0.518 | 0.509 | 0.510 | 0.521 | 0.515 | 0.524 | 0.514 | 0.519 |
| | MASF-D [30] | 0.505 | 0.505 | 0.504 | 0.508 | 0.505 | 0.491 | 0.516 | 0.504 | 0.518 | 0.507 |
| | MASF-P [30] | 0.502 | 0.501 | 0.499 | 0.517 | 0.505 | 0.491 | 0.510 | 0.493 | 0.524 | 0.504 |
| | Siamese Network [49] | 0.499 | 0.498 | 0.502 | 0.539 | 0.509 | 0.499 | 0.498 | 0.502 | 0.539 | 0.509 |
| | Reorder [104] | 0.548 | 0.542 | 0.530 | 0.568 | 0.547 | 0.567 | 0.564 | 0.552 | 0.571 | 0.563 |

Table 10: Model Performance of Repeated Depression Detection Using The Pre/Post-COVID Setup.

| Category | Model | Balanced Accuracy | | | ROC AUC | | |
|---|---|---|---|---|---|---|---|
| | | Pre-COVID | Post-COVID | Avg | Pre-COVID | Post-COVID | Avg |
| Baseline | Majority | 0.500 | 0.500 | 0.500 | 0.500 | 0.500 | 0.500 |
| Prior Depression Detection Model | Canzian et al. [18] | 0.495 | 0.500 | 0.497 | 0.479 | 0.490 | 0.484 |
| | Saeb et al. [78] | 0.515 | 0.524 | 0.519 | 0.519 | 0.534 | 0.526 |
| | Farhan et al. [33] | 0.481 | 0.519 | 0.500 | 0.495 | 0.537 | 0.516 |
| | Wahle et al. [91] | 0.529 | 0.523 | 0.526 | 0.531 | 0.532 | 0.531 |
| | Lu et al. [60] | 0.512 | 0.498 | 0.505 | 0.527 | 0.515 | 0.521 |
| | Wang et al. [97] | 0.513 | 0.534 | 0.524 | 0.536 | 0.545 | 0.541 |
| | Xu et al.-I [101] | 0.500 | 0.538 | 0.519 | 0.479 | 0.537 | 0.508 |
| | Xu et al.-P [102] | 0.511 | 0.505 | 0.508 | 0.533 | 0.505 | 0.519 |
| | Chikersal et al. [20] | 0.504 | 0.551 | 0.528 | 0.514 | 0.569 | 0.542 |
| Recent Domain Generalization Model | ERM-1dCNN [87] | 0.509 | 0.520 | 0.514 | 0.516 | 0.523 | 0.519 |
| | ERM-2dCNN [87] | 0.510 | 0.498 | 0.504 | 0.524 | 0.509 | 0.517 |
| | ERM-LSTM [87] | 0.515 | 0.510 | 0.512 | 0.515 | 0.511 | 0.513 |
| | ERM-Transformer [87] | 0.496 | 0.528 | 0.512 | 0.498 | 0.536 | 0.517 |
| | ERM-Mixup [111] | 0.503 | 0.511 | 0.507 | 0.498 | 0.513 | 0.506 |
| | IRM [7] | 0.499 | 0.498 | 0.499 | 0.501 | 0.501 | 0.501 |
| | DANN-D [39] | 0.514 | 0.513 | 0.514 | 0.515 | 0.530 | 0.522 |
| | DANN-P [39] | 0.500 | 0.500 | 0.500 | 0.490 | 0.507 | 0.499 |
| | CSD-D [72] | 0.506 | 0.518 | 0.512 | 0.511 | 0.524 | 0.517 |
| | CSD-P [72] | 0.516 | 0.515 | 0.516 | 0.520 | 0.518 | 0.519 |
| | MLDG-D [56] | 0.491 | 0.499 | 0.495 | 0.491 | 0.505 | 0.498 |
| | MLDG-P [56] | 0.503 | 0.497 | 0.500 | 0.508 | 0.509 | 0.509 |
| | MASF-D [30] | 0.496 | 0.511 | 0.504 | 0.498 | 0.522 | 0.510 |
| | MASF-P [30] | 0.498 | 0.519 | 0.509 | 0.503 | 0.525 | 0.514 |
| | Siamese Network [49] | 0.513 | 0.518 | 0.515 | 0.513 | 0.518 | 0.515 |
| | Reorder [104] | 0.523 | 0.528 | 0.525 | 0.536 | 0.542 | 0.539 |

Table 11: Model Performance of Repeated Depression Detection Using Overlapping Participants, using users in one dataset as the train set and the overlapping users in other datasets as the test set.

| Category | Model | Balanced Accuracy | | | | | ROC AUC | | | | |
|---|---|---|---|---|---|---|---|---|---|---|---|
| | | DS1 | DS2 | DS3 | DS4 | Avg | DS1 | DS2 | DS3 | DS4 | Avg |
| Baseline | Majority | 0.500 | 0.500 | 0.500 | 0.500 | 0.500 | 0.500 | 0.500 | 0.500 | 0.500 | 0.500 |
| Prior Depression Detection Model | Canzian et al. [18] | 0.571 | 0.500 | 0.494 | 0.420 | 0.496 | 0.570 | 0.361 | 0.343 | 0.429 | 0.425 |
| | Saeb et al. [78] | 0.626 | 0.624 | 0.463 | 0.547 | 0.565 | 0.658 | 0.685 | 0.330 | 0.582 | 0.564 |
| | Farhan et al. [33] | 0.460 | 0.500 | 0.455 | 0.503 | 0.480 | 0.421 | 0.593 | 0.431 | 0.529 | 0.494 |
| | Wahle et al. [91] | 0.536 | 0.500 | 0.479 | 0.532 | 0.512 | 0.559 | 0.627 | 0.394 | 0.560 | 0.535 |
| | Lu et al. [60] | 0.518 | 0.467 | 0.482 | 0.567 | 0.508 | 0.578 | 0.501 | 0.488 | 0.538 | 0.526 |
| | Wang et al. [97] | 0.603 | 0.500 | 0.475 | 0.548 | 0.532 | 0.620 | 0.500 | 0.493 | 0.617 | 0.557 |
| | Xu et al.-I [101] | 0.531 | 0.485 | 0.482 | 0.476 | 0.494 | 0.541 | 0.593 | 0.474 | 0.509 | 0.529 |
| | Xu et al.-P [102] | 0.548 | 0.548 | 0.560 | 0.518 | 0.544 | 0.555 | 0.571 | 0.602 | 0.539 | 0.567 |
| | Chikersal et al. [20] | 0.620 | 0.466 | 0.559 | 0.534 | 0.545 | 0.683 | 0.440 | 0.605 | 0.555 | 0.571 |
| Recent Domain Generalization Model | ERM-1dCNN [87] | 0.536 | 0.549 | 0.536 | 0.514 | 0.534 | 0.562 | 0.537 | 0.495 | 0.509 | 0.526 |
| | ERM-2dCNN [87] | 0.534 | 0.533 | 0.487 | 0.525 | 0.520 | 0.534 | 0.560 | 0.512 | 0.534 | 0.535 |
| | ERM-LSTM [87] | 0.514 | 0.546 | 0.475 | 0.567 | 0.525 | 0.513 | 0.546 | 0.461 | 0.601 | 0.530 |
| | ERM-Transformer [87] | 0.507 | 0.495 | 0.503 | 0.517 | 0.506 | 0.520 | 0.497 | 0.471 | 0.524 | 0.503 |
| | ERM-Mixup [111] | 0.536 | 0.549 | 0.536 | 0.514 | 0.534 | 0.562 | 0.537 | 0.495 | 0.509 | 0.526 |
| | IRM [7] | 0.534 | 0.525 | 0.468 | 0.504 | 0.508 | 0.564 | 0.530 | 0.445 | 0.555 | 0.524 |
| | DANN-D [39] | 0.469 | 0.522 | 0.467 | 0.471 | 0.482 | 0.464 | 0.523 | 0.486 | 0.508 | 0.495 |
| | DANN-P [39] | 0.435 | 0.507 | 0.500 | 0.500 | 0.486 | 0.441 | 0.509 | 0.459 | 0.477 | 0.472 |
| | CSD-D [72] | 0.539 | 0.534 | 0.443 | 0.553 | 0.517 | 0.567 | 0.562 | 0.423 | 0.590 | 0.535 |
| | CSD-P [72] | 0.512 | 0.578 | 0.443 | 0.525 | 0.515 | 0.519 | 0.610 | 0.430 | 0.544 | 0.526 |
| | MLDG-D [56] | 0.490 | 0.556 | 0.509 | 0.523 | 0.519 | 0.516 | 0.551 | 0.512 | 0.539 | 0.523 |
| | MLDG-P [56] | 0.499 | 0.539 | 0.472 | 0.534 | 0.511 | 0.516 | 0.552 | 0.469 | 0.535 | 0.518 |
| | MASF-D [30] | 0.567 | 0.547 | 0.501 | 0.513 | 0.532 | 0.576 | 0.565 | 0.494 | 0.524 | 0.540 |
| | MASF-P [30] | 0.560 | 0.510 | 0.525 | 0.526 | 0.530 | 0.545 | 0.529 | 0.517 | 0.528 | 0.530 |
| | Siamese Network [49] | 0.573 | 0.543 | 0.435 | 0.556 | 0.527 | 0.573 | 0.543 | 0.435 | 0.556 | 0.527 |
| | Reorder [104] | 0.614 | 0.633 | 0.532 | 0.513 | 0.573 | 0.673 | 0.699 | 0.526 | 0.517 | 0.604 |

# C  Dataset Statements & Documents

Our multi-year data collection study closely followed a sister study in Carnegie Mellon University (CMU). We acknowledge all efforts from CMU Study Team to provide important starting and reference materials. We state that we bear all responsibility in case of direct violation of participants' privacy right.

## C.1  Author Contribution Statement

We clarify every author's contribution to the datasets and the paper. Basic contributions like paper proof-reading are default and omitted. Leading conceptualization and effort are bolded.

- Xuhai Xu
  *Data Collection*: **Led technical parts of data collection in 2019 through 2021. Developed and maintained data collection applications from 2019 to 2021.** Assisted with data collection from 2019 to 2021; Assisted database maintenance of all years' datasets.
  *Analysis and Benchmark*: **Led curation of dataset, analysis, visualization, benchmarking, and data validation.** Main developer of benchmark platform GLOBEM.
  *Paper Writing & Supplementary Materials*: **Led paper writing, organization, and design of data sharing process.**

- Han Zhang
  *Data Collection*: **Developed and maintained data codebook and data cleaning (all years).** Assisted with the data collection from 2020 to 2021; quality assurance for data collection applications from 2019 to 2021.
  *Analysis and Benchmark*: **Led curation of dataset and visualization.** Assisted with analysis, benchmarking, and data validation.
  *Paper Writing & Supplementary Materials*: **Led curation of dataset details and data sharing agreement in supplementary materials.** Assisted with paper writing.

- Yasaman Sefidgar
  *Data Collection*: **Led design of infrastructure, pipeline and study codebase and codebooks impacting all years of data cleaning and processing; Led planning for 2019 data collection. Led technical parts of data collection in 2018 and 2019**. Also maintained database and study servers for 2018 and 2019; assisted with 2018 and 2019 data collection; and assisted with 2020 planning for data collection.
  *Analysis and Benchmark*: Not involved.
  *Paper Writing & Supplementary Materials*: Provided helpful comments.

- Yiyi Ren
  *Data Collection*: **Led transition of infrastructure for sensor data cleaning to RAPIDS**. Assisted with data collection study from 2019 - 2021; developed the study codebase, codebook and mobile applications that impact all years; maintained database and study servers from 2019 to 2021.
  *Analysis and Benchmark*: Not involved.
  *Paper Writing & Supplementary Materials*: Not involved

- Xin Liu
  *Data Collection*: Not involved.
  *Analysis and Benchmark*: Provided assistive effort with computing resources support, quality assurance, analysis, visualization, data validation, and GLOBEM development.
  *Paper Writing & Supplementary Materials*: Editing and framing.

- Woosuk Seo
  *Data Collection*: **Led 2018 data collection planning and data collection**. *Analysis and Benchmark*: Not involved.
  *Paper Writing & Supplementary Materials*: Not involved

- Jennifer Brown
  *Data Collection*: **Led 2020 data and 2021 data collection planning and data collection**. Assisted with codebook from 2019 to 2021.

- Kevin Kuehn
  *Data Collection*: **Led 2019 data collection planning and data collection**.

*Analysis and Benchmark*: Not involved.
*Paper Writing & Supplementary Materials*: Not involved

- Mike Merrill
  *Data Collection*: Not involved.
  *Analysis and Benchmark*: Assisted with data analysis, visualization, and benchmarking.
  *Paper Writing & Supplementary Materials*: Assisted with paper writing and study documentation in supplementary materials.

- Paula Nurius
  *Data Collection*: Supervised study material design and high-level planning for 2018-2021. Provided resources for study
  *Analysis and Benchmark*: Not involved.
  *Paper Writing & Supplementary Materials*: Supervised data-sharing agreement.

- Shwetak Patel
  *Data Collection*: Not involved.
  *Analysis and Benchmark*: Supervised data analysis. Provided computing resources.
  *Paper Writing & Supplementary Materials*: Editing.

- Tim Althoff
  *Data Collection*: Not involved.
  *Analysis and Benchmark*: Supervised data analysis, visualization, and benchmark results.
  *Paper Writing & Supplementary Materials*: Editing.

- Margaret E. Morris
  *Data Collection*: Supervised study material design and high-level planning for 2019-2021. Provided resources for study.
  *Analysis and Benchmark*: Not involved.
  *Paper Writing & Supplementary Materials*: Supervised data-sharing agreement. Editing.

- Eve Riskin
  *Data Collection*: Supervised study material design and high-level planning for 2018-2021. Provided resources for study.
  *Analysis and Benchmark*: Not involved.
  *Paper Writing & Supplementary Materials*: Supervised data-sharing agreement. Editing.

- Jennifer Mankoff
  *Data Collection*: Supervised study material design and high-level planning for 2018-2021. Provided resources for study.
  *Analysis and Benchmark*: Supervised data analysis and benchmark results.
  *Paper Writing & Supplementary Materials*: Supervised data-sharing agreement and paper writing.

- Anind K. Dey
  *Data Collection*: Supervised study material design and high-level planning for 2018-2021. Provided resources for study.
  *Analysis and Benchmark*: Supervised data analysis and benchmark results.
  *Paper Writing & Supplementary Materials*: Supervised data-sharing agreement and paper writing.

## C.2 Data Hosting, Licensing, and Maintenance Plan

Due to the sensitive nature of the dataset, we release our feature-level data with open credentialed access. Therefore, we plan to leverage the PhysioNet platform for data hosting and licensing, and maintenance.

**Host:** The PhysioNet platform with Credentialed Access.

**License:** PhysioNet Credentialed Health Data License 1.5.0

**Long-term Preservation:** PhysioNet is a well-known platform for freely-available health research data, software, challenges, and tutorials. It is a reliable platform for long-term hosting and preservation of our datasets. We will provide in-time maintenance for error correction through the platform. We will also actively maintain our benchmark platform GLOBEM.

### C.2.1 Dataset Meta-Data

We provide the meta-data below. Many texts are taken from the main paper. We adopt the meta-data format from PhysioNet as we will leverage it for the data release.

**Title:**

GLOBEM Dataset: Multi-Year Datasets for Longitudinal Human Behavior Modeling Generalization

**Abstract:**

We present the first multi-year mobile sensing datasets. Our multi-year data collection studies span four years (10 weeks each year, from 2018 to 2021). The four datasets contain data collected from 705 person-years (497 unique participants) with diverse racial, ability, and immigrant backgrounds. Each year, participants would install a mobile app on their phones and wear a fitness tracker. The app and wearable device passively track multiple sensor streams in the background $24\times7$, including location, phone usage, calls, Bluetooth, physical activity, and sleep behavior. In addition, participants completed weekly short surveys and two comprehensive surveys on health behaviors and symptoms, social well-being, emotional states, mental health, and other metrics. Our dataset analysis indicates that our datasets capture a wide range of daily human routines, and reveal insights between daily behaviors and important well-being metrics (*e.g.*, depression status). We envision our multi-year datasets can support the ML community in developing generalizable longitudinal behavior modeling algorithms.

**Background:**

Among various longitudinal sensor streams, smartphones and wearables are arguably one of the most widely available data sources [7]. The advances in mobile technology provide an unprecedented opportunity to capture multiple aspects of daily human behaviors, by collecting continuous sensor streams from these devices [10,11], together with metrics about health and well-being through self-report or clinical diagnosis as modeling targets. It poses unique challenges compared to traditional time-series classification tasks [6]. First, the data covers a much longer time period, usually across multiple months or years. Second, the nature of longitudinal collection often results in a high data missing rate. Third, the prediction target label is sparse, especially for mental well-being metrics.

Longitudinal human behavior modeling is an important multidisciplinary area spanning machine learning, psychology, human-computer interaction, and ubiquitous computing. Researchers have demonstrated the potential of using longitudinal mobile sensing data for behavior modeling in many applications, *e.g.*, detecting physical health issues [9], monitoring mental health status [11], measuring job performance [8], and tracing education outcomes [12]. Most existing research employed off-the-shelf ML algorithms and evaluated them on their private datasets. However, testing a model with new contexts and users is imperative to ensure its practical deployability. To the best of our knowledge, there has been no investigation of the cross-dataset generalizability of these longitudinal behavior models, nor an open testbed to evaluate and compare various modeling algorithms. To address this gap, we present the first multi-year passive mobile sensing datasets to help the ML community explore generalizable longitudinal behavior models.

**Methods & Technical Implementation:**

Our data collection studies were conducted at a Carnegie-classified R-1 university in the United State with an IRB review and approval. We recruited undergraduates via emails from 2018 to 2021. After the first year, previous-year participants were invited to join again. The study was conducted during Spring quarter for 10 weeks each year, so the impact of seasonal effects was controlled. Based on their compliance, participants received up to $245 in compensation every quarter.

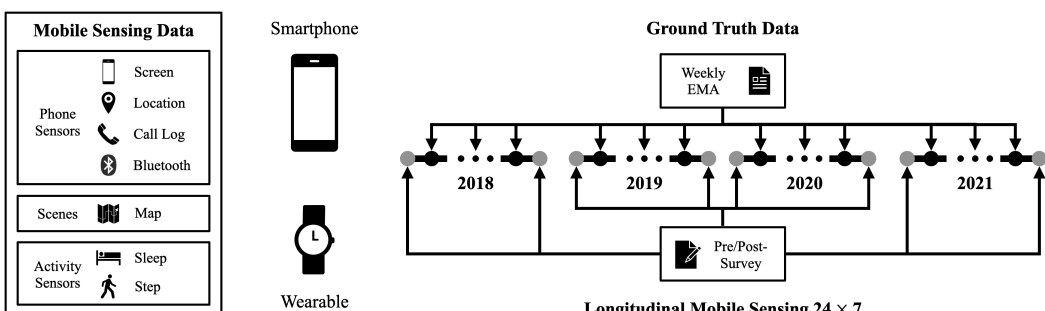

The four datasets (DS1 to DS4) have 155, 218, 137, and 195 participants (705 person-years overall, and 497 unique people). Our datasets have a high representation of females (58.9%), immigrants (24.2%), first-generations (38.2%), and disability (9.1%), and have a wide coverage of races, with Asian (53.9%) and White (31.9%) being dominant (*e.g.*, Hispanic/Latino 7.4%, Black/African American 3.3%).

*Part 1: Survey Data*

We collected survey data at multiple stages of the study. We delivered extensive surveys before the start and at the end of the study (pre/post surveys) and short weekly Ecological Momentary Assessment (EMA) surveys during the study to collect in-the-moment self-report data. All surveys consist of well-established and validated questionnaires to ensure data quality.

Our pre/post surveys include a number of questionnaires to cover various aspects of life, including 1) personality (BFI-10, The Big-Five Inventory-10), 2) physical health (CHIPS, Cohen-Hoberman Inventory of Physical Symptoms), 3) mental well-being (*e.g.*, BDI-II, Beck Depression Inventory-II; ERQ, Emotion Regulation Questionnaire), and 4) social well-being (*e.g.*, Sense of Social and Academic Fit Scale; EDS, Everyday Discrimination Scale). Our EMA surveys focus on capturing participants' recent sense of their mental health, including PHQ-4, Patient Health Questionnaire 4; PSS-4, Perceived Stress Scale 4; and PANAS, Positive and Negative Affect Schedule.

We use the depression detection task as a starting point for behavior modeling. We employ BDI-II (post) and PHQ-4 (EMA) as the ground truth. Both are screening tools for further inquiry of clinical depression diagnosis. We focus on a binary classification problem to distinguish whether participants' scores indicate at least mild depressive symptoms through the scales (*i.e.*, PHQ-4 > 2, BDI-II > 13). The average number of depression labels is 11.6±2.6 per person. The percentage of participants with at least mild depression is 39.8±2.7% for BDI-II and 46.2±2.5% for PHQ-4.

Due to some design iteration, we did not include PHQ-4 in DS1, but only PANAS. Although PANAS contains questions related to depressive symptoms (*e.g.*, "distressed"), it does not have a comparable theoretical foundation for depression detection like PHQ-4 or BDI-II. Therefore, to maximize the compatibility of the datasets, we trained a small ML model on DS2 that has both PANAS and PHQ-4 scores to generate reliable ground truth labels. Specifically, we used a decision tree (depth=2) to take PNANS scores on two affect questions ("depressed" and "nervous") as the input and predict PHQ-4 score-based depression binary label. Our model achieved 74.5% and 76.3% for accuracy and F1-score on a 5-fold cross-validation on DS2. The rule from the decision tree is simple: the user would be labeled as having no depression when the distress score is less than 2, and the nervous score is less than 3 (on a 1-5 Likert Scale). We then applied this rule to DS1 to generate depression labels.

*Part 2: Sensor Data*

We developed a mobile app using the AWARE Framework [5] that continuously collects location, phone usage (screen status), Bluetooth scans, and call logs. The app is compatible with both the iOS and Android platforms. Participants installed the app on smartphones and left it running in the background. In addition, we provided wearable Fitbits to collect their physical activities and sleep

behaviors. The mobile app and wearable passively collected sensor data 24×7 during the study. The average number of days per person per year is 77.5±8.9 among the four datasets.

**Content description:**

We release four datasets, named `INS-W_1`, `INS-W_2`, `INS-W_3`, and `INS-W_4`. A dataset has three folders. We provided an overview description below. Please refer to our GitHub README page for more details.

- `SurveyData`: a list of files containing participants' survey responses, including pre/post long surveys and weekly short EMA surveys.
- `FeatureData`: behavior feature vectors from all data types, using RAPIDS [2] as the feature extraction tool.
- `ParticipantInfoData`: some additional information about participants, *e.g.*,, device platform (iOS or Android).

Specifically, the folder structure of a dataset folder is shown as follows:

`root of a dataset folder`

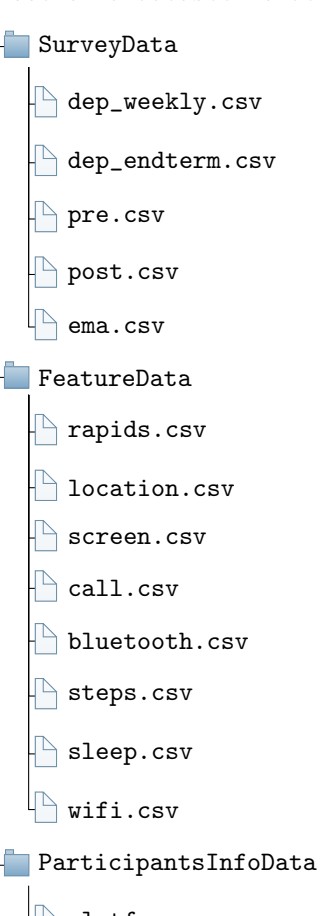

```
SurveyData
    dep_weekly.csv
    dep_endterm.csv
    pre.csv
    post.csv
    ema.csv
FeatureData
    rapids.csv
    location.csv
    screen.csv
    call.csv
    bluetooth.csv
    steps.csv
    sleep.csv
    wifi.csv
ParticipantsInfoData
    platform.csv
```

The `SurveyData` folder contains five files, all indexed by `pid` and `date`:

- `dep_weekly.csv`: The specific file for depression labels (column `dep`) combining post and EMA surveys. We also have PHQ4 sub-scales as column `dep_weekly_subscale` for the depression sub-scale and `anx_weekly_subscale` for the anxiety sub-scale. Moreover, we have another column merging post and PHQ-4 depression sub-scale at `dep_weeklysubscale_endterm_merged`.
- `dep_endterm.csv`: The specific file for depression labels (column `dep`) only in post surveys. Some prior depression detection tasks focus on end-of-term depression prediction.

These two files are created for depression as it is the benchmark task. We envision future work can be extended to other modeling targets as well.

- `pre.csv`: The file contains all questionnaires that participants filled in right before the start of the data collection study (thus pre-study).
- `post.csv`: The file contains all questionnaires that participants filled in right after the end of the data collection study (thus post-study).
- `ema.csv`: The file contains all EMA surveys that participants filled in during the study. Some EMAs were delivered on Wednesdays, while some were delivered on Sundays.

PS: Due to the design iteration, some questionnaires are not available in all studies. Moreover, some questionnaires have different versions across years. We clarify them using column names. For example, INS-W_2 only has `CESD_9items_POST`, while others have `CESD_10items_POST`. `CESD_9items_POST` is also calculated in other datasets to make the modeling target comparable across datasets.

The `FeatureData` folder contains seven files, all indexed by `pid` and `date`.

- `rapids.csv`: The complete feature file that contains all features.
- `location.csv`: The feature file that contains all location features.
- `screen.csv`: The feature file that contains all phone usage features.
- `call.csv`: The feature file that contains all call features.
- `bluetooth.csv`: The feature file that contains all Bluetooth features.
- `steps.csv`: The feature file that contains all physical activity features.
- `sleep.csv`: The feature file that contains all sleep features.
- `wifi.csv`: The feature file that contains all WiFi features. Note that this feature type is not used by any existing algorithms and often has a high data missing rate.

Please note that all features are extracted with multiple `time_segments`

- morning (6 am - 12 pm, calculated daily)
- afternoon (12 pm - 6 pm, calculated daily)
- evening (6 pm - 12 am, calculated daily)
- night (12 am - 6 am, calculated daily)
- allday (24 hrs from 12 am to 11:59 pm, calculated daily)
- 7-day history (calculated daily)
- 14-day history (calculated daily)
- weekdays (calculated once per week on Friday)
- weekend (calculated once per week on Sunday)

For all features with numeric values, we also provide two more versions:

- normalized: subtracted by each participant's median and divided by the 5-95 quantile range
- discretized: low/medium/high split by 33/66 quantile of each participant's feature value

The `ParticipantInfoData` folder contains files with additional information.

- `platform.csv`: The file contains each participant's major smartphone platform (iOS or Android), indexed by `pid`
- `demographics.csv`: Due to privacy concerns, demographic data are only available for special requests. Please reach out to us directly with a clear research plan with demographic data.

**Usage notes:**

We provide a behavior modeling benchmark platform GLOBEM [1]. The platform is designed to support researchers in using, developing, and evaluating different longitudinal behavior modeling methods.

Researchers who use the datasets must agree to the following terms.

**Commercial use** The database will not be used for non-academic research purposes. Non-academic purposes include but are not limited to:

- proving the efficiency of commercial systems
- training or testing of commercial systems
- using screenshots of subjects from the dataset in advertisements
- selling data from the dataset
- creating military applications
- developing governmental systems used in public spaces

**Distribution** The database will not be re-distributed, published, copied, or further disseminated in any way or form whatsoever, whether for profit or not. This includes further distributing, copying or disseminating to a different facility or organizational unit in the requesting university, organization, or company, with the exception of using small portions of data for the exclusive purpose of clarifying academic publications or presentations.

**Privacy** Although the database has been anonymized, we cannot eliminate all potential risks of privacy information leakage. The PI of any research group access to the dataset, is responsible for continuing to safeguard this database, taking whatever steps are appropriate to protect participants' privacy and data confidentiality. The specific actions required to safeguard the data may change over time.

**Misuse** If at any point, the administrators of the datasets at the University of Washington have concerns or reasonable suspicions that the researcher has violated these usage note, the researcher will be notified. Concerns about misuse may be shared with PhysioNet and other related entities.

Our datasets have led to multiple publications:

- Sefidgar YS, Seo W, Kuehn KS, Althoff T, Browning A, Riskin E, Nurius PS, Dey AK, Mankoff J. Passively-sensed behavioral correlates of discrimination events in college students. Proceedings of the ACM on human-computer interaction. 2019 Nov 7;3(CSCW):1-29.
- Zhang H, Nurius P, Sefidgar Y, Morris M, Balasubramanian S, Brown J, Dey AK, Kuehn K, Riskin E, Xu X, Mankoff J. How does COVID-19 impact students with disabilities/health concerns?. arXiv preprint arXiv:2005.05438. 2020 May 11.
- Xu X, Chikersal P, Dutcher JM, Sefidgar YS, Seo W, Tumminia MJ, Villalba DK, Cohen S, Creswell KG, Creswell JD, Doryab A. Leveraging Collaborative-Filtering for Personalized Behavior Modeling: A Case Study of Depression Detection among College Students. Proceedings of the ACM on Interactive, Mobile, Wearable and Ubiquitous Technologies. 2021 Mar 29;5(1):1-27.
- Nurius PS, Sefidgar YS, Kuehn KS, Jung J, Zhang H, Figueira O, Riskin EA, Dey AK, Mankoff JC. Distress among undergraduates: Marginality, stressors and resilience resources. Journal of American college health. 2021 May 30:1-9.
- Morris ME, Kuehn KS, Brown J, Nurius PS, Zhang H, Sefidgar YS, Xu X, Riskin EA, Dey AK, Consolvo S, Mankoff JC. College from home during COVID-19: A mixed-methods study of heterogeneous experiences. PloS one. 2021 Jun 28;16(6):e0251580.
- Sefidgar YS, Nurius PS, Baughan A, Elkin LA, Dey AK, Riskin E, Mankoff J, Morris ME. Examining Needs and Opportunities for Supporting Students Who Experience Discrimination. arXiv preprint arXiv:2111.13266. 2021 Nov 25.
- Xu X, Mankoff J, Dey AK. Understanding practices and needs of researchers in human state modeling by passive mobile sensing. CCF Transactions on Pervasive Computing and Interaction. 2021 Dec;3(4):344-66.

There are a few known limitations in these datasets:

- Limited study population: Only a portion of students who received our emails or social media posts would participate in our study, which could only represent a subset of the general population.
- High data missing rate: Missing data is inevitable due to various reasons, such as low battery, data transfer loss, and sensor permission withdrawal. Survey data can also be missing sometimes due to the lack of compliance.

**Ethics:**

Our datasets aim at aiding research efforts in the area of developing, testing, and evaluating machine learning algorithms to better understand college students' (and potentially more general population) daily behaviors, health, and well-being from continuous sensor streams and self-reports. These findings may support public interest in how to improve student experiences and drive policy around adverse events students and others may experience.

Privacy is the major ethical concern of our data collection studies. We strictly follow the IRB rules to anonymize participants' data. Anyone outside our core data collection group cannot access direct individually-identifiable information. We also eliminated the data for users who stopped their participation at any time during the study. Since some sensitive sensor data (*e.g.*, location) can disclose identities, we only release feature-level data under credentialing to protect against privacy leakage.

**Data collected from human subjects:**

The study protocol was approved by relevant Institutional Review Boards (IRBs). Human participants signed a consent form before participating in the study.

**Clinical trial data:** N/A

**Data collected from animals:** N/A

**Acknowledgments:**

Our multi-year data collection study closely followed a sister study at Carnegie Mellon University (CMU). We acknowledge all efforts from CMU Study Team to provide important starting and reference materials [4]. Moreover, our studies were greatly inspired by StudentLife researchers from Dartmouth College [3].

Our studies were supported by the University of Washington (including the Paul G. Allen School of Computer Science and Engineering; Department of Electrical and Computer Engineering; Population Health; Addictions, Drug and Alcohol Institute; and the Center for Research and Education on Accessible Technology and Experiences); the National Science Foundation (EDA-2009977, CHS-2016365, CHS-1941537, IIS1816687 and IIS7974751), the National Institute on Disability, Independent Living and Rehabilitation Research (90DPGE0003-01), Samsung Research America, and Google

**Conflicts of interest:**

The author(s) have no conflicts of interest to declare.

**Version:** 1.0.0

**References & Weblinks:**

[1] Benchmark Platform GLOBEM. github.com/UW-EXP/GLOBEM

[2] Rapids documentation. https://www.rapids.science/1.6/.

[3] Studentlife study https://studentlife.cs.dartmouth.edu/, 2014.

[4] A. Doryab, D. K. Villalba, P. Chikersal, J. M. Dutcher, M. Tumminia, X. Liu, S. Cohen, K. Creswell, J. Mankoff, J. D. Creswell, et al. Identifying behavioral phenotypes of loneliness and social isolation with passive sensing: statistical analysis, data mining and machine learning of smartphone and fitbit data. JMIR mHealth and uHealth, 7(7):e13209, 2019.

[5] D. Ferreira, V. Kostakos, and A. K. Dey. Aware: Mobile context instrumentation framework. Frontiers in ICT, 2:6, 2015.

[6] R. Godahewa, C. Bergmeir, G. I. Webb, R. J. Hyndman, and P. Montero-Manso. Monash time series forecasting archive. In Neural Information Processing Systems Track on Datasets and Benchmarks, 2021.

[7] N. D. Lane, E. Miluzzo, H. Lu, D. Peebles, T. Choudhury, and A. T. Campbell. A survey of mobile phone sensing. IEEE Communications Magazine, 48(9), 2010.

[8] S. M. Mattingly, J. M. Gregg, P. Audia, A. E. Bayraktaroglu, A. T. Campbell, N. V. Chawla, V. Das Swain, M. De Choudhury, S. K. D'Mello, A. K. Dey, et al. The tesserae project: Large-scale, longitudinal, in-situ, multimodal sensing of information workers. In Extended Abstracts of the 2019 CHI Conference on Human Factors in Computing Systems, pages 1–8, 2019.

[9] J.-K. Min, A. Doryab, J. Wiese, S. Amini, J. Zimmerman, and J. I. Hong. Toss "n" turn: Smartphone as sleep and sleep quality detector. In Proceedings of the SIGCHI Conference on Human Factors in Computing Systems, CHI '14, page 477–486, New York, NY, USA, 2014. Association for Computing Machinery.

[10] M. E. Morris, Q. Kathawala, T. K. Leen, E. E. Gorenstein, F. Guilak, W. DeLeeuw, and M. Labhard. Mobile therapy: case study evaluations of a cell phone application for emotional self-awareness. Journal of medical Internet research, 12(2):e10, 2010.

[11] R. Wang, F. Chen, Z. Chen, T. Li, G. Harari, S. Tignor, X. Zhou, D. Ben-Zeev, and A. T. Campbell. Studentlife: Assessing mental health, academic performance and behavioral trends of college students using smartphones. In Proceedings of the 2014 ACM International Joint Conference on Pervasive and Ubiquitous Computing, pages 3–14. ACM, 2014.

[12] R. Wang, G. Harari, P. Hao, X. Zhou, and A. T. Campbell. Smartgpa: how smartphones can assess and predict academic performance of college students. In Proceedings of the 2015 ACM international joint conference on pervasive and ubiquitous computing, pages 295–306, 2015.

