# OpenReview forum: "GLOBEM Dataset: Multi-Year Datasets for Longitudinal Human Behavior Modeling Generalization"
_NeurIPS.cc/2022/Track/Datasets_and_Benchmarks — NeurIPS 2022 Datasets and Benchmarks _

### Official Review · Reviewer_qRCD · 2022-07-11
**A rich dataset for studying depression**

**Rating:** 6
**Confidence:** 3

**Strengths:**

- Previous benchmarks in depression prediction use audio/video data (AVEC, DAIC-WOZ), text (Reece et al.), survey data (Sau et al) or biomarkers(Sharma et al) ; it seems thus that this dataset complements existing benchmarks. The Depresjon dataset has an activity feature and is the most similar but lacks rich features.
- The dataset collection is well documented (compensation, IRB notice, forms distributed to study participants etc)
- Rich collection of results including semi-manual/psychology backed methods and ML methods
- The multi-year nature and pre/post-covid seem to be very useful to benchmark OOD methods

**Weaknesses:**

- Table 1 lacks the actual metric being used (r/Model Performance/Balanced Accuracy/ probably)
- Even though it is difficult and costly the dataset size could be increased as 700 instances is a bit low for benchmarking ML models with a risk to overfit quickly the test set
- The self-labeling is debatable - but this has been correctly stated by the authors
- it is not clear to me how useful is the dataset as it seems that among 18 methods not one is able to perform (eg in ODD DomainBed has several datasets with >>50% Accuracy) - could it be that features are not that predictive ?
- (main weakness #1) It is not clearly stated what model selection method was used to choose hyperparameters of the different methods :(
- In OOD Gulrajani et al have observed that leave-one-domain out was not the best model selection approach so it could be interesting to provide results using training domain validation, leave one domain out and oracle selection (see the DomainBed paper).
- (main weakness #2) The WOODs benchmark (https://woods-benchmarks.github.io/) is widely used in ODD and presents as well sensor time series data (even though the label is different): the difference should be discussed in the paper

**Additional Feedback:**

I appreciate the way authors have provided access to their dataset and code.

**Clarity:**

- mostly yes - only the caption of Tab1 is notvery clear and a definition of balanced accuracy would be welcome in text

**Correctness:**

- mostly yes - it is unfortunately unclear how hyperparam search was conducted ie which model selection method was used

**Documentation:**

yes
it could be interesting that the authors provide screenshots of the application

**Ethics:**

I think one area of ethical discussion that's missing (or assumed positive w/o much discussion) is the usefulness for the society to predict depression from sensor data. If the answer is yes it could lead to debatable usage such as targeting people with predicted mental health issues to influence their behavior in a manner that is not necessarily positive for them. For instance with the intent sell them medications that are not necessarily effective (see eg a recent article in the MIT tech review about debunked cancer treatment ads on Facebook).

As such I believe there should be a more thorough discussion about the relevance for the society to ease development of mental health prediction methods based on widely available sensor data.

**Relation To Prior Work:**

- yes (see Strengths)

**Summary And Contributions:**

The dataset encompasses several original features:
- multi-year (thus providing a realistic task for generalization)
- rich implicit data from sensors, inc. time series
- relatively diverse population (yet all of them are students)

The paper also provides
- results for 18 possible models
- complete information on dataset collection, discussion of privacy concerns

Part of the dataset has already been used to publish (eg. Doryab et al).

---

> ### Author Response · Authors · 2022-08-12
> **Response to qRCD**
>
> We thank the reviewer for the great suggestions. Please find our response below.
>
> - Metric Clarification
> > A definition of balanced accuracy would be welcome in text
>
> We updated Table 2's caption (originally Table 1) to clarify that balanced accuracy was used as the metric. In addition, we added the definition of balanced accuracy in Sec. 5.3.
>
> - Algorithm Performance
> > it is not clear to me how useful is the dataset as it seems that among 18 methods not one is able to perform. could it be that features are not that predictive?
>
> We would like to highlight the difference between behavior modeling tasks and computer vision tasks. In DomainBed, the datasets involved are all image-based data, which has different formats and patterns from longitudinal passive sensing data. Therefore, the results are not comparable.
>
> Our results are indeed comparable to prior work focusing on the depression detection task (although their dataset is not publicly available). For example, [99] achieved an ROC AUC of 0.759 on DS3 (0.724 on average), while [95] achieved an ROC AUC of 0.809 on their own dataset. These results indicate that our features are predictive, but the depression detection task is challenging. (On a separate note, [95] performed poorly on our datasets -- best ROC AUC 0.667 on DS4, but 0.567 on average -- which indicates the lack of generalizability of [95] on other datasets.) We added a new paragraph in Sec.6 about potential directions to improve model generalizability. We look forward to future contributions from the community to further improve the modeling performance.
>
> - Model Selection Method (main #1)
> > It is not clearly stated what model selection method was used to choose hyperparameters
>
> For ML models, we implemented a built-in grid search process while training the models to pick the one with the best performance using cross-validation within the training datasets.
>
> For DL models, we did not adopt a grid search as it is overly computationally expensive. We simply adopted default values for many hyperparameters (these values can be found in the config folder of our Github codebase). For example, we pick Adam as the optimizer as this is commonly used, and a learning rate of 0.001 is a common value for Adam. Same with the cosine annealing schedule and some other parameters. For the architecture, we conducted some simple experiments with the layer size (8, 16, 32) and depth (3,5,7), which led to similar results. Thus we adopted the smaller ones (layer size as 8, depth as 3) to reduce computing costs. We added these clarifications in S.B.2.2.
>
> It is worth noting that the model selection is not the major contribution of this work. We envision that future researchers can conduct further model selection processes on our platform.
>
> - Other Benchmark and Datasets (main #2)
> > The WOODs benchmark is widely used in ODD and presents as well sensor time series data: the difference should be discussed in the paper
> > Even though it is difficult and costly the dataset size could be increased as 700 instances is a bit low for benchmarking ML models with a risk to overfit quickly the test set
>
> We thank the reviewer for pointing us to WOODs. We agree that it is a recent well-established benchmark for OOD time-series data. However, we would like to highlight that existing datasets covered in WOODs are different from our datasets which include longitudinal passive mobile sensing datasets for human behavior. We discussed the uniqueness of our work in Sec. 2, and also added the reference of WOODs in several places in the paper.
>
> Moreover, compared to other depression prediction work using passive mobile sensing datasets, our datasets contain much larger sample sizes.
>
> We added a new Table 1 in Sec. 2 to compare our multi-year datasets against other related datasets (including WOODs) and highlight 1) the uniqueness of our work, and 2) the large sample size of our datasets.
>
> - Additional Study Details
> > it could be interesting that the authors provide screenshots of the application.
>
> We provide the instruction slides we present to participants when they joined the study, which include the screenshots of the application. We followed the reviewer's suggestion and added the screenshot into a new section S.A.3.
>
>
> - More Ethical Discussion
> > There should be a more thorough discussion about the relevance for the society
>
> We moved the ethical discussion part from Sec. 6 to Sec. 5.5 and extended the short paragraph with more discussion on the purpose of doing such a modeling task.

---

> ### Author Response · Authors · 2022-08-26
> **Follow up message**
>
> Dear Reviewer qRCD,
>
> We hope that our response has adequately addressed your questions and concerns. If there is any further question, please feel free to let us know any time. Thank you again for your time and valuable feedback!

---

### Official Review · Reviewer_6DfS · 2022-07-24
**Extensive multi-year datasets containing mobile sensing data and well-being metrics**

**Rating:** 7
**Confidence:** 4

**Strengths:**

- This study focuses on in-the-wild time-series data which has not been widely explored in terms of domain generalization compared to computer vision or natural language processing domains.
- They collect sufficient features: location, phone usage, Bluetooth, call, physical activity, and sleep. Participants completed weekly short surveys and extensive surveys before the start and at the end of the study. Pre/post surveys cover personality (BFI-10), physical health (CHIPS), mental well-being (BDI-II), and social well-being(Sense of Social and Academic Fit Scale; EDS). The short surveys include PHQ-4, PSS-4, and PANAS.
- This covers data from four consecutive years which is extensive enough to be served as longitudinal research. The four years consist of two years before COVID and two years after COVID. This leads to a domain shift between datasets, and it makes the released data more informative and suitable for targeting domain generalization.
- The benchmark contains both depression detection algorithms and domain generalization algorithms.

**Weaknesses:**

- It would have been better if they provide the statistics of their data clearly and compare it to the few other datasets which can be compared to GLOBEM dataset. Also, as they mentioned, mobile sensing data tend to have a high data missing rate, but they did not provide the full missing rate of each feature. It is critical for time-series data.
- This is targeting human behavior modeling, but the benchmark covers a depression detection task alone with one label. It would be better to at least suggest some other tasks related to human behavior modeling with this extensive dataset.
- It is limited to a certain population such as students, Asian and White, and immigrants and first-generation participants.
- The results of the cross dataset setup represent the domain generalizability. Domain classification tasks are provided to show the domain shift of their dataset; the “Name-The-Dataset” experiment addresses the domain shift in the Leave-One-Dataset-Out setup, and the “Distinguish-The-Person” experiment explains the domain shift in the “Overlapping Users across Datasets”. Section 4.1 partly explains the domain shift in the Pre/Post-COVID setup, but it would be better to add similar experiment.

**Additional Feedback:**

- About Figure 6(a), after feature normalization, ‘Year1’ scores 25% which is the same as random guessing. I wonder if we still can tell that there is a domain shift between ‘Year1’ and the others.
- There are numerous survey data, and to my knowledge, only BDI-II and PHQ-4 are used in this study. I would like to know more examples of human behavior modeling other than the task provided. I also wonder if all the survey data provided for the reviewers will be released publicly.
- Several domain generalization algorithms with different strategies (i.e. data manipulation, representation learning, and learning strategy) are covered. I would like to know how the authors selected nine methods and ask if they also consider other well-known domain generalization methods such as groupDRO, VRex, and JTT.

**Clarity:**

- This paper is well written containing an introduction, background, dataset explanation, dataset analysis, benchmark, discussion, and conclusion.
- The authors clearly explain the dataset collection procedure and the details of the dataset with adequate figures.

**Correctness:**

- The dataset is constructed in a sound way. For the benchmark, they provide a balanced accuracy as a metric, but they also show a ROC AUC in supplementary material.
- Considering that depression detection is a binary classification task, the performances of the baselines seem to be quite low even in a single dataset setup. Also, I wonder why they only include traditional ML models for depression detection algorithms and the majority of deep learning models are based on CNN even though the dataset is time-series data.

**Documentation:**

They provide sufficient detail on data collection and organization, availability and maintenance, and ethical and responsible use. The materials are well-provided for the reviewers as well. The readme file in the link given by the authors details the reproduction of their benchmark.

**Ethics:**

- They strictly follow the IRB rules to anonymize participants’ data. In order to prevent privacy leakage, only feature-level data is released.
- However, it contains the exact dates of the collected data, so it would have been better if it shifts the time as well in a similar way to publicly available electronic health records.

**Relation To Prior Work:**

They clearly discuss that this work is different from other datasets targeting domain generalization in terms of the modality. Also, they explain that this dataset is differentiated from other time-series mobile sensing data in that it covers multi-years.

**Summary And Contributions:**

- The authors release the first multi-year longitudinal mobile sensing datasets, covering over 700 person-years multiple sensor datasets (i.e. location, phone usage, calls, Bluetooth, physical activity, and sleep behavior) and well-being metrics (e.g. physical health, mental well-being, social well-being). Well-being metrics are collected by pre/post surveys and short twice-weekly Ecological Momentary Assessment surveys during the study. Their benchmark results on the depression detection task demonstrate that current algorithms need to be further developed in terms of generalizability.
- They analyze the distribution of data before and after COVID or on weekdays and weekends. They also conduct correlation analysis, capturing the relationship between daily behaviors and well-being metrics. The two domain classification tasks they conduct show significant distribution shifts among datasets and individuals.
- They provide benchmark results of 9 prior depression detection algorithms and 9 deep-learning domain generalization algorithms on the depression detection task. The experimental setups are divided into a single dataset and a cross dataset.
- The best depression detection algorithms are better at the single-dataset task, while the best domain generalization algorithms are better at cross-dataset tasks. A significant performance drop from the single dataset task to the three cross-datasets shows that current algorithms are still far from real-life deployment.

---

> ### Author Response · Authors · 2022-08-12
> **Response to 6DfS**
>
> We thank the reviewer for the helpful feedback. Please find our response below.
>
> - Comparison with Other Datasets
> > It would have been better if they provide the statistics of their data clearly and compare it to the few other datasets which can be compared to GLOBEM dataset.
>
> This is a constructive suggestion. We added a new Table 1 in Sec. 2 to compare our multi-year datasets against other related datasets.
>
> - Missing Rate of All Features
> > They did not provide the full missing rate of each feature. It is critical for time-series data.
>
> We agree with the reviewer that reporting the missing rate is important. We added the missing rate information in the feature detail table in S.A.7 . We also highlighted it in Sec. 3.3.
>
> - Envision of Extra Behavior Modeling Tasks
> > It would be better to at least suggest some other tasks related to human behavior modeling with this extensive dataset.
>
> We thank the reviewer for the comment. We added a new paragraph in Sec. 6 to envision potential new behavior modeling tasks besides depression detection.
>
> - Dataset Limitation
> > It is limited to a certain population such as students, Asian and White, and immigrants and first-generation participants.
>
> We thank the reviewer for pointing this out. We agree with the reviewer that our datasets have restrictions on population. Although this population does not include all age groups, we focused on those most at risk of mental health issues (according to [National Institutes of Health](https://www.nimh.nih.gov/health/statistics/mental-illness#:~:text=Prevalence%20of%20Any%20Mental%20Illness%20(AMI),-Figure%201%20shows&text=The%20prevalence%20of%20AMI%20was,50%20and%20older%20(14.5%25).)). Furthermore, our datasets are not limited to Asian and White, but also other racial groups, as well as immigrants and first-generation participants. Instead, we intentionally oversampled minoritized groups (such as underrepresented minorities) to make our datasets more representative. We highlight this in Sec. 3.1.
>
> - More Domain Classification Analysis
> > Section 4.1 partly explains the domain shift in the Pre/Post-COVID setup, but it would be better to add a similar experiment.
> > ‘Year1’ scores 25% which is the same as random guessing. I wonder if we still can tell that there is a domain shift between ‘Year1’ and the others.
>
> We agree with the reviewer that Year1’s score (slightly above 25%) indicates that DS1's domain shift can be reduced significantly after feature normalization. We claim that in Sec. 4.3 (lines 247-248). As for pre/post-COVID classification analysis, the pre-COVID period corresponds to Year 1&2 (DS1&2), and the post-COVID period corresponds to Year 3&4 (DS3&4). Thus, our "Name-the-Dataset" already covers this task.
>
> - Choice of Algorithms
> > why they only include traditional ML models for depression detection algorithms and the majority of deep learning models are based on CNN even though the dataset is time-series data.
>
> The vast majority of existing passive sensing-based depression detection studies used small datasets collected over several weeks or months (e.g., [21,95,99]). There are very few studies leveraging deep learning models on longitudinal behavior data. To make our results more comparable, we decided to start with existing ML algorithms.  As for using CNN as the backbone, we experimented with common architectures such as LSTM and Transformer, which under-performed CNNs. We speculated this could be explained by the four-week data format (28 days), which can be easy for CNN to capture global information. Moreover, our datasets enable future researchers to test with more ML/DL algorithms and architectures for the depression detection task and other behavior modeling tasks.
>
> > how the authors selected nine methods and ask if they also consider other well-known domain generalization methods.
>
> To answer the question of why we chose these domain generalization algorithms. We picked them to cover the high-level taxonomy: 1) data manipulation (Mixup), 2) representation learning (IRM, DANN, CSD), and 3) learning strategy (MLDG, MASF, Siamese, Reorder). We added the clarification to S.B.2.2. We agree with the reviewer that other well-known algorithms are also worth being tested on our datasets, and we envision that our datasets can enable the ML community to experiment, test, and compare more algorithms.
>
> - Release of Dataset
> > I also wonder if all the survey data provided for the reviewers will be released publicly.
>
> All data of the surveys listed in S.A.6 will be released together with the depression detection tasks in `SurveyData/ema.csv, pre.csv, post.csv`.
>
> > it would have been better if it shifts the time as well in a similar way to publicly available electronic health records
>
> We appreciate the reviewer's great suggestion on date shifting. We will shift the dates by the time we publish our datasets.

---

> ### Author Response · Authors · 2022-08-26
> **Follow up message**
>
> Dear Reviewer 6DfS,
>
> We would like to thank you again for your time and positive, valuable feedback!
>
> We hope that our response has adequately addressed your questions and concerns. There are a few days before the end of the discussion period. If you have any further questions or suggestions, please feel free to let us know anytime!

---

> > ### Comment · Reviewer_6DfS · 2022-09-01
> > **Response to review**
> >
> > I apologize for the late response.
> > However, I read your review as soon as you replied and it sufficed all of my concerns.
> > I am glad that I gave some helpful suggestions to you.
> > I appreciate that you take some of my suggestions and supplement your paper.
> > Hope you can get a good result!

---

### Official Review · Reviewer_aRHQ · 2022-07-24
**Review for GLOBEM Paper**

**Rating:** 5
**Confidence:** 4
**Correctness:** Yes
**Clarity:** Yes

**Strengths:**

** The surveys and the depression detection task are well performed.

** The paper is well written and the features for the depression detection task are interesting.

** The dataset seems promising for mental health future research


**Weaknesses:**

** What is the different between dataset in [7] and this dataset since the authors are the same?


** The dataset associated with the submission then is not fully open as claimed in the Introduction


** A short background of the depression detection problem and the related algorithms would make the paper more readable.


** Big missing rate for the data may affect generalizability of the findings since a trivial method is used for handling such missing data


** In page 6, having a rigor analysis for each year and compare between Pandemic and Post-pandemic is more interesting compared to just a general correlation score?


** In page 8, the authors state "Current algorithms’ cross-dataset generalizability is still far from satisfactory for real-life deployment". A more detailed explaining of the intuition of such challenge and suggested directions are essential for the paper, despite having short paragraph in Discussion section.


** Having the self-report aspect for depression measures and not having backup intelligent prediction is also a shortcoming of the paper.


** The following sentence is not clear "Our datasets can serve as an open testbed for multiple cross-dataset generalization tasks to evaluate a behavior modeling algorithm’s generalizability and robustness (e.g., same or different users across multiple years)".


** The following finding seems trivial, and it is well known from psychology literature "This suggests that participants often commuted to off-campus places on weekends. Further, participants tended to leverage weekends to catch up on sleep, as shown by the peak in-bed duration around weekends".


**Additional Feedback:**

More details of the depression detection task and making the dataset fully public will be helpful

**Documentation:**

Not perfect

**Ethics:**

Yes

**Relation To Prior Work:**

Yes

**Summary And Contributions:**

The paper presents the first multi-year passive sensing datasets, containing over 700 users’ data collected from mobile and wearable sensors, together with a wide range of well-being metrics. They benchmarked several algorithms to develop generalizable longitudinal behavior modeling algorithms with focusing on depression detection task.

---

> ### Author Response · Authors · 2022-08-12
> **Response to aRHQ**
>
> We thank the reviewer for the important and insightful feedback. Please find our response below.
>
> - Relationship with The Other Submission
> > What is the different between dataset in [7] and this dataset since the authors are the same?
>
> We thank the reviewer for pointing this out. Although both papers leveraged the GLOBEM platform, their contributions are very different.
>
> The major contribution of [7] (a journal paper targeted at the ubiquitous computing and HCI community) is proposing a new behavior modeling algorithm called Reorder that has a consistent yet marginal improvement on behavior model’s generalizability.
>
> However, the major contribution of this NeurIPS paper includes: 1. releasing our large-scale, multi-user datasets to support the ML community in developing generalizable longitudinal behavior modeling algorithms, 2. reporting benchmarks of multiple existing algorithms on the depression detection task.
>
> The authors of [7] and the authors of this paper have some overlap but they are not the same. [7] used different datasets and did not consider the dataset release. Moreover, [7] used a small set of behavior features and did not explore pre/post-COVID comparison.
>
> In order to provide full transparency for reviewers, we included an anonymized version of [7] in the supplementary materials. Please do not share [7] as it is still under a double-blind review process. Thanks!
>
> - Background of Depression Detection
> > A short background of the depression detection problem and the related algorithms would make the paper more readable.
>
> We followed the reviewer's suggestion and added a brief background introduction of depression detection and algorithms in Sec. 5.
>
> - Data Imputation Methods
> > Big missing rate for the data may affect generalizability of the findings
>
> We agree with the reviewer that a better imputation technique might be able to improve the modeling results. Our current benchmark results serve as a starting point with a naive imputation technique, and we look forward to having future researchers design more imputation techniques on our datasets. We added more discussion in Sec. 6 about this.
>
> - Pre/post-COVID Correlation Analysis
> > In page 6, compare between Pandemic and Post-pandemic is more interesting
>
> We see values of both types of analysis. The four-year analysis can reveal consistent and potentially generalizable feature correlations, while the pre/post-COVID analysis can reveal the impact of COVID. We added a new part in S.A.5 with the pre/post COVID analysis and discussion suggested by the reviewer.
>
> - More Discussion about Potential Directions to Improve Generalizability
> > A more detailed explaining of the intuition of such challenge and suggested directions are essential
>
> We appreciate the reviewer's suggestions and also agree that discussing possible directions is important. We added a new paragraph in Sec. 6 to discuss potential future direction.
>
> - Debatable Self-report Labels
> > Having the self-report aspect for depression measures and not having backup intelligent prediction is also a shortcoming of the paper.
>
> Self-reporting via clinically validated questionnaires is one of the diagnosis steps in nowadays state-of-the-art methods of measuring depression and other mental health issues. It also has the advantage that frequent labels are relatively easier to obtain in the wild. The more strict procedure also involves clinician interview and fMRI, but the high cost of time/effort and the social stigma makes the labels challenging to collect. We further added extra discussion in Sec. 6 about this limitation.
>
> - Claim Updates
>
> > The sentence is not clear.
>
> We clarified the sentence in Sec. 1 (around line 60) to be "Our datasets can serve as an open testbed for multiple cross-dataset generalization tasks (e.g., same users different years, different users same years, different users different years) to evaluate a behavior modeling algorithm's generalizability and robustness".
>
>
> > The finding seems trivial
>
> In response to the reviewer’s comments on our findings of weekly behavior patterns, we agree that these findings are intuitive. We state them in order to show that our datasets can capture a wide range of daily routines of participants. We shortened this part in Sec. 4.1 and merged this paragraph into the previous paragraph.
>
> > The dataset associated with the submission then is not fully open as claimed in the Introduction
>
> Our S.A.6 and S.A.7 detail the data we are planning to release. We are happy to provide more details if helpful.

---

> ### Author Response · Authors · 2022-08-26
> **Follow up message**
>
> Dear Reviewer aRHQ, thank you again for your time and valuable feedback!
>
> We hope that our response has adequately addressed your questions and concerns. Is there anything else we can answer for you?

---

### Official Review · Reviewer_EUnZ · 2022-07-27
**Reviews of GLOBEM**

**Rating:** 6
**Confidence:** 3
**Clarity:** Yes.

**Strengths:**

1. The provided dataset is the first multi-year datasets from mobile and wearable sensing over 700 person-years with overlapping users. Interesting, the dataset includes pre/post COVID behavioral data.

2. The data collection process is clear, which shows the rarity and input cost. Besides, this paper provides relevant data analysis in terms of distribution, correlation, and domain classification.

3. The authors conduct experiments of two tasks (i.e. Prior Depression Detection and Recent Domain Generalization), which offer a useful benchmark result for future research.

**Weaknesses:**

1. Inadequate sample size and type. Although the dataset contains four years of behavioral data, the number of users is small and the number of behavior types collected is also small.

2. Lack of comparison with other human behavior datasets, which is a very important work for dataset papers.

3. I encourage the authors to provide a discussion of the effects of errors introduced by the collection equipment, experimental participants.

**Additional Feedback:**

None.

**Correctness:**

The collection of the dataset seems to be canonical. But there are two doubts here.
1. Why is there the operation "Weekly EMA" in Fig?
2. Are the collection results from different devices accurate enough? Is there a possible effect of different years of collection equipment on experimental results?

**Documentation:**

The dataset is available.

**Ethics:**

There may be ethical issues with this paper, but they are discussed in lines 331-336.

**Relation To Prior Work:**

A comparison of the relevant behavioral datasets can provide is appropriate.

**Summary And Contributions:**

This paper provides the first longitudinal mobile and wearable sensing dataset, and reports the benchmark results for 18 related algorithms for the depression detection. Besides, in this paper, the data collection process and visualized analysis(e.g. data distribution and correlation)  are present, which help researchers to understand this dataset quickly.

---

> ### Author Response · Authors · 2022-08-12
> **Response to EUnZ**
>
> We thank the reviewer for the helpful feedback. Please find our response below.
>
> - Comparison with Other Datasets
> > Lack of comparison with other human behavior datasets, which is a very important work for dataset papers.
> > The number of users is small and the number of behavior types collected is also small.
>
> We thank the reviewer for pointing this out, to address this concern, we added a new table (i.e., Table 1) in Sec. 2 to compare our multi-year datasets against other related datasets. The comparison indicates that the size of our datasets is significantly larger than existing open passive mobile sensing datasets.
>
> - Potential Study Errors
> > I encourage the authors to provide a discussion of the effects of errors.
>
> To ensure the robustness of our apps and devices, we conducted pilot tests across multiple devices before each year's formal data collection. We also developed a data dashboard to ensure that participants’ data collection app was functioning properly during the study. However, we did acknowledge that device differences (e.g., different OS and smartphone models) and human errors (e.g., battery drain and network issue) may introduce potential bias to the data. We added this information to Sec. 6, S.A.3, and S.A.4.
>
> - Weekly EMA Operation in Fig. 1
> > Why is there the operation "Weekly EMA" in Fig?
>
> Each year, our label data came from three periods: at the beginning of the study (pre), during the study (weekly EMA), and at the end of the study (post). Our depression detection task combined the weekly EMA (PHQ-4) and post-survey labels (BDI-II).

---

> ### Author Response · Authors · 2022-08-26
> **Follow up message**
>
> Dear Reviewer EUnZ,
>
> We hope that our response has adequately addressed your questions and concerns. If you have any further questions, please feel free to let us know any time. Thank you again for your time and great feedback!

---

### Official Review · Reviewer_JD92 · 2022-07-27
**The first multi-year behavior dataset**

**Rating:** 7
**Confidence:** 3
**Correctness:** The evaluation methods and experiment…
**Clarity:** The paper is well written and easy to…

**Strengths:**

1. The first multi-year behavior dataset that supports domain generalization study
2. Thorough and comprehensive data collection/processing description
3. An out-of-the-box benchmarking platform with 18 integrated algorithms

**Weaknesses:**

My main concern is the possible overlapping content with the authors' under-review submission [7]. It seems these two papers use the same platform, i.e., GLOBEM. The authors also claim that "Many texts are taken from [7] with courtesy".

**Additional Feedback:**

The authors could list the scales of (open-source/private) datasets used in previous studies so that the readers can assess if 700+ users are enough.

**Documentation:**

The algorithms and evaluation pipelines are accessible on GitHub with the detailed README. There seems to be sufficient detail to support reproducibility.

**Ethics:**

The major ethical concern on privacy has been carefully tackled by the authors.

**Relation To Prior Work:**

The authors clearly discuss how this work differs from previous papers in Background section.  But the authors should clarify how this paper relates to their under-review submission [7].

**Summary And Contributions:**

The authors focus on modeling longitudinal behavior signals and observe that there are no publicly available datasets with multiple domains. So the authors propose the first multi-year mobile and wearable sensing datasets as an open testbed. The authors re-implement 18 methods and provide the benchmark results. Extensive results reveal the limitations of the current methods, e.g., lack of generalizability. The dataset along with the benchmark results will advance the development of generalizable longitudinal behavior modeling.

---

> ### Author Response · Authors · 2022-08-12
> **Response to JD92**
>
> We thank the reviewer for the meaningful suggestions. Please find our response below.
>
> - Relationship with The Other Submission
> > My main concern is the possible overlapping content with the authors' under-review submission [7]
>
> We thank the reviewer for pointing this out. Although both papers leveraged the GLOBEM platform, their contributions are very different.
>
> The major contribution of [7] (a journal paper targeted at the ubiquitous computing and HCI community) is proposing a new behavior modeling algorithm called Reorder that has a consistent yet marginal improvement on behavior model’s generalizability.
>
> However, the major contribution of this NeurIPS paper includes: 1. releasing our large-scale, multi-user datasets to support the ML community in developing generalizable longitudinal behavior modeling algorithms, 2. reporting benchmarks of multiple existing algorithms on the depression detection task.
>
> The authors of [7] and the authors of this paper have some overlap but they are not the same. [7] used different datasets and did not consider the dataset release. Moreover, [7] used a small set of behavior features and did not explore pre/post-COVID comparison.
>
> In order to provide full transparency for reviewers, we included an anonymized version of [7] in the supplementary materials. Please do not share [7] as it is still under a double-blind review process. Thanks!
>
> - Comparison with Other Datasets
> > The authors could list the scales of (open-source/private) datasets used in previous studies so that the readers can assess if 700+ users are enough.
>
> We thank the reviewer for the great suggestion. We added a new table (i.e. Table 1) in Sec. 2 to compare the sample size of our datasets against previous datasets. The comparison indicates that our datasets contain significantly larger sample sizes than other datasets.

---

> ### Author Response · Authors · 2022-08-26
> **Follow up message**
>
> Dear Reviewer JD92,
>
> We hope that our response has adequately addressed your questions and concerns. If you have any further questions, please feel free to let us know! Thank you again for your time and feedback. We appreciate it!

---

### Official Review · Reviewer_8Nxe · 2022-07-28
**GLOBEM: Multi-Year Datasets for Longitudinal Human Behavior Modeling Generalization**

**Rating:** 8
**Confidence:** 3
**Clarity:** The paper is generally well written.

**Strengths:**

It is an impressive experimental procedure to collect data over the course of 4 years from hundreds of subjects.  Having consistent data from such periods can be beneficial for a variety of machine learning purposes.  The included mental health information provides one immediate use, but hopefully the data can be leveraged for other purposes as well.

The presented graphs provide good visualizations of the data, clearly demonstrating weekly trends and differentiating the datasets.  The correlations results indicating how selected metrics relate to the depressions results was also quite interesting.

A wide variety of machine learning pipelines are also tested.

The data is presented as CSV files, which is a highly portable format.

**Weaknesses:**

A few questions/comments that came to mind are summarized below:

- It is mentioned that the platform may be extended to other behavior modeling tasks beyond depression detection, but it is unclear how ground-truth labels would be obtained for other tasks.  I may suggest adding some discussion about the possible scope of future tasks and how labels would be created for them.  This may be worth also addressing a bit in the abstract (the scope of "behavior modeling"), introduction, etc. if possible .

- It may be helpful to clarify/elaborate the phrase "cross-dataset" that is prominent in the discussions.  It is intuitive to the extent that it considers applying a model across different datasets, but scope and the definition of "dataset" could be clarified.  For example, GLOBEM seems to be considered as 4 datasets where each one is from a different year, but I would tend to consider it as a single multi-year dataset.  In addition, the scope of cross-dataset analysis in this case keeps constant all of the data streams, formats, targets, experimental protocols, etc.  This seems like it may be more aptly considered as splitting a dataset into multiple years, especially with overlapping subjects in each split, whereas cross-dataset sounds like there may be more substantial differences or approaches.  Note that this ties nicely with your points about it being a challenging task, since even this case where the datasets are so similar is still challenging.

- It is great to explore so many different models, but I wonder about their target domain.  Were each of them designed for a task such as this using the same data streams available in this dataset?  If they were designed to operate for somewhat different tasks, it may help explain their relatively low performance.  It was odd that they were all so low even on single-dataset conditions, so I wonder where their strengths lie and how their original applications differ from the current tasks.

- Given the relatively low performance of all algorithms, I wasn't entirely sure of the desired conclusions.  Is it simply meant to show that this is a hard task?  Is it meant as a benchmark for future depressions detection pipelines?  This is also related to the previous question about how the training data and experimental procedures of the tested pipelines compare to the current tests, to ensure that it is a fair test for them.  It was mentioned that they suffer from overfitting, but their scores on single-dataset tasks were also fairly low.

- The distinguish-a-person task seems to actually be distinguishing person-years rather than unique subjects, since there are 705 targets instead of 497.  Maybe this causes issues with the training seeing more data from some subjects than others, and conflating subject differences with dataset differences?  Some additional discussion may be interesting here.

- Just to check, are the domain generalization algorithms simply different network architectures that were designed to generalize well, or is there a more fundamental difference between them and the depression detection algorithms?

- The abstract mentions over 700 users, which may be misleading since there are 497 unique users.  (Of course, both numbers are very impressive!)

- Until I looked at some data files, it wasn't immediately clear that the datasets' sampling rate was effectively once per day (that summary data is provided for each stream for each day).  I know it was mentioned on line 185, but it seems notable enough to highlight it.  I'm not quite sure where the aggregated features from the other time ranges are presented, but perhaps they are in different data files.

- Some more information about RAPIDS may be helpful to include in the main paper.

- It may be helpful to mention the Fitbit models used.  Presumably they didn't include heart rate or user-initiated exercise labels?

**Additional Feedback:**

Congrats on an impressive multi-year data collection study!!

**Correctness:**

The experimental protocols sound reasonable, and the data files seem clearly organized.  Regarding evaluation methods/benchmarks, see the few comments above about adding discussion to address how the data, training, and experiments of the existing models compare to the current test scenarios to ensure it is a fair comparison.

**Documentation:**

The GitHub seems to have thorough documentation and instructions.  I don't think the long-term storage/hosting plan was discussed yet, but presumably it would be after publication due to the potentially sensitive nature.

**Ethics:**

The paper mentions key privacy concerns and ethical considerations, but more discussions could be provided about ethical considerations for using the dataset (it may be in the supplemental materials, but including it in the paper and in the data distribution website would be helpful).

**Relation To Prior Work:**

The advantages of spanning multiple years and many users are clearly discussed.  Making a subsection of section 2 explicitly about related datasets may be beneficial.

**Summary And Contributions:**

The paper summarizes a dataset of smartphone and Fitbit data spanning 439 unique subjects across 4 10-week periods (one period per year).  It contains data about phone screen usage, location, call durations, nearby Bluetooth devices, maps, sleep, and steps.  It also includes surveys related to mental health and wellbeing. Statistics and distributions are presented to summarize the data, visualizing weekly trends and comparisons between pre-COVID and post-COVID data.  ML models are used to distinguish datasets, distinguish subjects, predict depression, and generalize across datasets.  A variety of existing learning pipelines are tested.

---

> ### Author Response · Authors · 2022-08-12
> **Response to 8Nxe**
>
> We thank the reviewer for the positive comments and helpful suggestions. Please find our response below.
>
> - Envision of Extra Behavior Modeling Tasks
> > I may suggest adding some discussion about the possible scope of future tasks
>
> We thank the reviewer for the constructive suggestion. We modified Sec. 1 to specify the motivation of picking depression detection as our starting point (lines 64-66). In addition, we added a new paragraph in Sec. 6 to envision potential new behavior modeling tasks besides depression detection (lines 368-377), and modified the conclusion to adjust the scope (lines 391-392).
>
> - Clarification of Cross-datasets
> > It may be helpful to clarify/elaborate the phrase "cross-dataset" that is prominent in the discussions.
>
> In this paper, generalization would include the modeling of 1) different users collected from the same years (same dataset), 2) same users collected from different years (cross-dataset), and 3) different users collected from different years (cross-dataset). We clarified the term of “cross-dataset” in Sec. 1 (lines 60-61).
>
> - Choice of Algorithms
> > If they were designed to operate for somewhat different tasks, it may help explain their relatively low performance.
>
> All ML algorithms proposed in previous literature focus on the depression detection task. However, the domain generalization algorithms are not specifically designed for longitudinal passive sensing data, but mostly for CV or NLP tasks. We agree with the reviewer that this may explain the low performance. We clarified it in Sec. 6.
>
> > is there a more fundamental difference between them and the depression detection algorithms
>
> We picked these algorithms to cover the high-level taxonomy: 1) data manipulation (Mixup), 2) representation learning (IRM, DANN, CSD), and 3) learning strategy (MLDG, MASF, Siamese, Reorder). We added the clarification to S.B.2.2. We agree with the reviewer that other well-known algorithms are also worth being tested on our datasets, and we envision that our datasets can enable the ML community to experiment, test, and compare more algorithms.
>
> - The Conclusion from Benchmark Results
> > Given the relatively low performance of all algorithms, I wasn't entirely sure of the desired conclusions.
>
> Longitudinal behavior modeling is not an easy task given the noisy behavior of humans in nature. Although our benchmarks are not very high, our results can serve as a starting point for future researchers to further explore. We added a paragraph in Sec. 6 to discuss the potential future directions to improve model generalizability.
>
> - Clarification of Dataset Details
>
> > The abstract mentions over 700 users, which may be misleading
>
> We thank the reviewer for pointing this out. To clear the ambiguity,  we added the number of unique participants in the abstract.
>
> > It wasn't immediately clear that the datasets' sampling rate was effectively once per day
> > I'm not quite sure where the aggregated features from the other time ranges are presented
>
> As for the feature name and sampling rate, we employed a specific naming format of all features: `[feature_type]:[feature_name][_norm or NULL]:[time_segment]`. For example, given a feature "timeathome", we will have "f_loc:timeathome:allday" for 24 hours daily feature value, "f_loc:timeathome:morning" for the value calculated in the morning episode, and "f_loc:timeathome:14dhist" for the value calculated from the past two weeks. All these features can be calculated on a daily basis. We added additional clarification in S.A.7.
>
> > Some more information about RAPIDS may be helpful to include in the main paper.
> > It may be helpful to mention the Fitbit models used. Presumably they didn't include heart rate or user-initiated exercise labels?
>
> For RAPIDS details, we added more descriptions of RAPIDS in Sec. 3.3. For Fitbit, we used Fitbit Flex2 and Inspire2 in our study. Both models do not include heart rate data. We did not collect user-initiated exercise labels. We added more clarification in S.A.3.
>
> > I don't think the long-term storage/hosting plan was discussed yet
>
> For the dataset maintenance plan, we plan to resort to the PhysioNet platform to host our dataset with credentialed access. S.C.2 lists out the meta-data information we plan to put on the PhysioNet once the paper is accepted.
>
> - Comparison with Other Datasets
>
> > Making a subsection of section 2 explicitly about related datasets may be beneficial.
>
> We thank the reviewer for the great suggestion. We added a new table (i.e., Table 1) in Sec. 2 to compare our multi-year datasets against other related datasets.
>
> - More Ethics Discussion
>
> > More discussions could be provided about ethical considerations for using the dataset
>
> We thank the reviewer for providing this helpful comment. We moved the ethical discussion part from Sec. 6 to Sec. 5.5 and extended the short paragraph with more discussion on the purpose of doing such a modeling task.

---

> > ### Comment · Reviewer_8Nxe · 2022-09-02
> > **Response to responses**
> >
> > Thank you for the thoughtful responses and revision!

---

> ### Author Response · Authors · 2022-08-26
> **Follow up message**
>
> Dear Reviewer 8Nxe,
>
> We would like to thank you again for your time and positive, constructive feedback!
>
> We hope that our response has adequately addressed your questions and concerns. There are a few days before the end of the discussion period. If you have any further questions, please feel free to let us know anytime!

---

### Official Review · Reviewer_Z7SD · 2022-07-28
**The dataset has great potential to the field and it should be accepted after some issues be addressed.**

**Rating:** 7
**Confidence:** 4
**Clarity:** Yes.

**Strengths:**

- The contribution is significant, and the dataset is relevant to the broader research community. The collected dataset includes a large scale of data (smartphones, wearables, multiple questionnaires) from 705 person-years. Various behavioral modeling algorithms for depression detection tasks are evaluated on the dataset and show the lack of generalizability of all existing algorithms. This dataset may assist researchers to develop more generalizable algorithms and explore interesting insights into the relationship between sensing data and human well-being metrics.
- The accessibility and accountability are good. All the data are clearly presented and explained.
- The ethics are well considered, and the documents related to data collection are complete.


**Weaknesses:**

1. The authors claimed that one of the contributions is that the dataset can serve as an open testbed for cross-dataset generalization tasks to evaluate the generalizability and robustness of behavioral modeling algorithms. However, there are many public datasets online and they can also serve as the open testbed for evaluation. If one advantage of this dataset is the capability of evaluating generalization, why not compare the depression detection performance of existing algorithms on other popular datasets?
2. Why do you use binary classification? I think doing regression may more accurately reflect the depression score, especially considering the relatively low accuracy of the experiment result.
3. Where is S.A.5, I could not find the feature details as S.A.5 is missing.
4. Please explain the reason for every decision you made, e.g., why add a 3-level discretized version instead of 2 or 4-level? In addition, you mention that you omit the missing values during analysis and use a median-based imputation when necessary. What about the features with a very high missing rate? In this case, median-based imputation may be not accurate.
5. Correlation analysis. In the manuscript, Pearson correlation coefficients were computed between every feature and the depression label. However, the calculation of the p-value relies on the assumption that each item is normally distributed. I could not find the description of the distribution of features.
6. Technical validation. Authors should provide information to justify the reliability of their data (e.g., statistical analysis of experimental error and variation, discussion of any procedures used to ensure reliable and unbiased data production).



**Additional Feedback:**

N/A

**Correctness:**

Yes. The dataset is constructed in a reasonable way, but each decision made needs more explanation.

**Documentation:**

Yes. The documents are sufficient.

**Ethics:**

Yes, ethics are well considered in the research.

**Relation To Prior Work:**

Yes, this work compares the studentlife study and explains how the work differs from the previous study. However, I am not convinced by the claim that ‘the only public longitudinal time-series sensor dataset is from the pioneering StudentLife study’. As far as I know, there are some other longitudinal time-series sensor datasets not mentioned. It would be good to add a table to compare the related datasets and clearly show how the dataset differs from others.

[1] Wang R, Aung M S H, Abdullah S, et al. CrossCheck: toward passive sensing and detection of mental health changes in people with schizophrenia[C]//Proceedings of the 2016 ACM international joint conference on pervasive and ubiquitous computing. 2016: 886-897.

[2] Gao N, Marschall M, Burry J, et al. Understanding occupants’ behaviour, engagement, emotion, and comfort indoors with heterogeneous sensors and wearables[J]. Scientific Data, 2022, 9(1): 1-16.


**Summary And Contributions:**

This manuscript presents a longitudinal mobile and wearable sensing dataset for four years, with various types of self-report data such as personality, physical health, depression, emotion and social well-being. It also lists the benchmark results of 18 algorithms for depression detection tasks and supports cross-dataset evaluations of behavior modelling algorithms’ generalizability. Although many human behavioural modelling research were done and some of related datasets were released previously, there is no open dataset for comparing and evaluating various modelling algorithms, bring difficulty to the development of this field. In this research, the dataset is very well presented, and the experiments are carefully designed. I believe it has great potential to the field and it should be accepted with some issues addressed.

---

> ### Author Response · Authors · 2022-08-12
> **Response to Z7SD**
>
> We thank the reviewer for the important comments. Please find our response below.
>
> - Comparison with Other Datasets
> > Why not compare the depression detection performance of existing algorithms on other popular datasets?
>
> We agree with the reviewer that comparing algorithms' performance on other public datasets (the ones that support depression detection in Table 1) can further reveal more insights into these algorithms. However, enabling a consistent feature extraction pipeline for heterogeneous datasets collected from different research groups is not a trivial task (RAPIDS only supported data collected from a limited set of smartphone apps and wearable devices). We envision this as one of the potential future studies that can extend our current datasets and further advance the generalizability research in this area.
>
> > There are some other longitudinal time-series sensor datasets not mentioned. It would be good to add a table to compare the related datasets and clearly show how the dataset differs from others.
>
> We thank the reviewer for pointing us to additional datasets. We added a new table (i.e., Table 1) in Sec. 2 to list related datasets and compare our multi-year datasets against them.
>
> - Classification vs. Regression
> > Why do you use binary classification?
>
> We combined PHQ-4 (in weekly EMAs) and BDI-II (in post-survey). Since their scales are at different ranges, they cannot support the regression task. However, they have the same depression distinction levels (4-level: none or minimal, mild, moderate, and severe), which enables classification tasks. We picked binary classification as a starting point and envision future work to explore multi-class classification. Moreover, when only using PHQ-4 or BDI-II, we agree with the reviewer that regression is another feasible modeling task. This can be potential future exploration on our datasets.
>
> - Missing Supplementary Details
> > Where is S.A.5, I could not find the feature details as S.A.5 is missing.
>
> After adding more details, the latest feature details are in S.A.7. Please find them in the supplementary material folder.
>
> - Dataset Decision
> > Why add a 3-level discretized version instead of 2 or 4-level?
>
> We explain our decision of selecting feature time ranges and post-processing as follows. For feature time ranges, we picked them based on previous studies in the ubicomp community [19, 23], including four epochs of a day, all day, past one week, and past two weeks.
>
> As for post-processing, we provided normalization and discretization. Normalization is a commonly used technique among ML tasks. And 3-level discretization was conducted based on prior work such as [99]. It is worth noting that researchers are not limited to these two post-processing techniques. We are releasing the feature data so that researchers can develop their own post-processing techniques to support their purposes.
>
> - Data Imputation Methods
> > Median-based imputation may be not accurate.
>
> We agree with the reviewer that a median-based imputation could be overly naive. Imputation of such longitudinal passive sensing data can be an important research topic itself and goes beyond the scope of this dataset and benchmark paper. We added more discussion about this as future work in Sec. 6.
>
> - Correlation Analysis
> > The calculation of the p-value relies on the assumption that each item is normally distributed.
>
> We appreciate the reviewer's helpful comment. We replaced Pearson correlation analysis with Spearman correlation, which does not have the assumption of the normal distribution when calculating p-values. We updated Fig. 5 with the new results. Notably, the claim in the paper remains the same.
>
> - Data Validation
> > Authors should provide information to justify the reliability of their data
>
> Before each year's study, we tested our data collection app on multiple smartphone brands to ensure its compatibility, robustness, and data collection quality. We also developed a data dashboard to ensure that participants’ data collection app was functioning properly during the study. We added these details in a new section S.A.3.

---

> ### Author Response · Authors · 2022-08-26
> **Follow up message**
>
> Dear Reviewer Z7SD,
>
> Thank you again for your time and positive, constructive feedback!
>
> We hope that our response has adequately addressed your questions and concerns. There are a few days before the end of the discussion period. If you have any further question, please feel free to let us know any time!

---

### Review · Ethics_Reviewer_u7iw · 2022-08-21

**Recommendation:** 1

**Ethics Documentation:**

- The data collection and data preparation is detailed in the README.md file.
- The dataset is anonymized. The presence of location information of the patients raises privacy concerns, but the authors address this concern in the main paper.
- Data sharing and maintenance plan is elaborated in the supplementary material.

It would be valuable to add the Ethical considerations or maintenance plan in the README of the github repo, or at least clearly point to the section of the paper where these details are provided.



**Ethics Review:**

No major ethical concerns that the authors do not address in the submission.

---

> ### Author Response · Authors · 2022-08-21
> **Response to u7iw - Ethics Review**
>
> We thank the reviewer for your time and comments.
>
> In the revised version of our paper (please find the new version by downloading the latest version of the pdf), we have added a new section (Sec 5.5) to incorporate more discussion about ethical concerns.
>
> We will also make sure to include Ethical discussion in the github repo and data website when we publish our codebase and datasets.

---

### Official Review · Reviewer_wDSv · 2022-09-13
**Valuable multi-year passive sensing dataset for human behavior modeling**

**Rating:** 9
**Confidence:** 3
**Clarity:** The paper is well written and easy to…

**Strengths:**

- The dataset is a valuable resource for comparing algorithms used for modeling human behavior or predicting certain human characteristics.
- The documentation is quite elaborate and easy to follow


**Weaknesses:**

The performance of the algorithms for the detection task is not very impressive. There may be possible noises and hopefully curating more data over years would make it better.

**Additional Feedback:**

I recommend the paper to be accepted

**Correctness:**

The techniques used for curating the data and comparing the benchmarks are sound.

**Documentation:**

The documentation is detailed and contain sufficient information on data collection, organization, and maintenance.

**Ethics:**

No ethical concerns

**Relation To Prior Work:**

The paper positions itself very well in comparison to the existing datasets available around longitudinal phone and sensor datasets.

**Summary And Contributions:**

The paper curates data from longitudinal mobile and wearable sensors over multiple years. The dataset contains additional information on physical health, depression, etc. The paper also demonstrates how the dataset can be used to compare benchmark algorithms aimed at depression detection tasks. The paper performs cross-dataset evaluations of the benchmark algorithms.

---

### Author Response · Authors · 2022-08-12
**Overall Summary of Paper Revision**

We thank all reviewers for their thoughtful and insightful comments. We appreciate the recognition of the paper's contributions. According to reviewers, our paper has an “impressive” (8Nxe), “well-performed” study (aRHQ) and “significant” contribution (Z7SD), and provides an “extensive” dataset (6DfS, JD92) and “useful” benchmark results (qRCD, EUnZ).

We also received a lot of constructive feedback and suggestions from reviewers on how to further improve the paper. We responded to each reviewer individually. Here we summarize the overall list of revisions we have made to the paper, with the order of paper sections.

We updated the submission with a clean revised version. In the supplementary material, we also provide a version with all changes highlighted in green color (named `GLOBEM_full_highlighted.pdf`).

- Abstract
  - Line 8: We clarified the number of unique participants (8Nxe)
- Sec. 1: Introduction
  - Line 60: We clarified the definition of cross-dataset task (8Nxe, aRHQ)
  - Line 65: We added a short envision of other potential behavior modeling tasks. We elaborated on more details in the discussion (6DfS, 8Nxe).
- Sec. 2: Background
  - Line 95: We added the recent benchmark WOODS (qRCD)
  - Line 110: We incorporated more related datasets (Z7SD) and added a new Table 1 to compare our dataset against other related public or private datasets, which indicates that our datasets have a larger scale than prior datasets (qRCD, 6DfS, EUnZ, JD92, 8Nxe, Z7SD)
- Sec. 3: Datasets
  - Line 127: We highlighted the diverse population representations in our datasets (6DfS).
  - Line 165: We added more information about RAPIDS (8Nxe).
  - Line 197: We added an additional high-level claim about the missing rate. Please refer to S.A.7 for more details (6DfS).
- Sec. 4 Analysis
  - Line 211: We shorten the discussion of intuitive weekly pattern findings (aRHQ).
  - Line 229: We replaced Pearson correlation with Spearman correlation and updated the text and Figure 5 (Z7SD).
- Sec. 5 Benchmark
  - Line 258: We added more background information about depression detection (aRHQ).
  - Line 274: We added more background introduction about related algorithms for depression detection in the ubiquitous computing and HCI community (aRHQ).
  - Line 208: We added the definition of balanced accuracy and updated Table 2’s caption accordingly (qRCD).
  - Line 324: We moved the original short paragraph of ethics from Sec. 6 to Sec. 5.5, and extend it significantly (qRCD, 8Nxe).
- Sec. 6 Discussion & Conclusion
  - Line 352: We added additional discussion of the poor algorithm performance (8Nxe).
  - Line 356: We added a new paragraph to discuss some promising directions to improve model generalizability (qRCD, aRHQ, 8Nxe).
  - Line 367: We added a new paragraph to discuss other potential behavior modeling tasks that can be conducted on our extensive datasets (6DfS, 8Nxe).
  - Line 378: We modified the limitation discussion significantly to incorporate debatable self-report labels (aRHQ), potential errors from devices and participants (EUnZ), and imputation methods (aRHQ, Z7SD).
  - Line 389: We clarified the scope of our paper (8Nxe).
- Supp. A
  - Line 716: We added a new S.A.3 to introduce more details of study hardware (qRCD), as well as study setups to ensure the quality of our data collection studies (EUnZ, 8Nxe, Z7SD).
  - Line 747: We added more discussion of the potential bias introduced by devices (EUnZ).
  - Line 751: We added a new S.A.5 to incorporate additional correlation analysis to compare pre- vs. post-COVID periods (aRHQ).
  - Line 769: We added more clarification of feature details (8Nxe) and missing rate information (6DfS).
- Supp. B
  - Line 848: We added an overview of the three categories covered by domain generalization techniques (6DfS, 8Nxe).
  - Line 898: We added more details about hyperparameter tuning (qRCD).

We thank all reviewers again for their thoughtful reviews. We believe our paper is greatly improved as a result of these suggestions. We look forward to your response!

---

### Meta-Review · Area_Chair_Xdt6 · 2022-09-10

**Recommendation:** Accept
**Confidence:** 5

**Metareview:**

GLOBEM is a multi-year human behaviour sensing dataset captured from mobile and wearable devices. This is a highly impactful dataset that will benefit several research domains including machine learning, HCI and ubiquitous computing, affective computing, and mental health research. The reviewers largely agree on the strong contributions, both on the dataset as well as the benchmark task on domain generalisation induced from the behaviour before and after COVID-19.
Although several techniques that are adopted, such as the imputation techniques, are on the naive side, this can open opportunities for others to work on the dataset and improve the modelling aspects.
Therefore, I strongly recommend acceptance of the work.

---

### Decision · Program_Chairs · 2022-09-16

Accept